# EXPO-GS: EXPOSURE-AWARE SIGNED DISTANCE FUNCTION IN GAUSSIAN SPLATTING FOR HIGH DYNAMIC RANGE

## ABSTRACT

High dynamic range novel view synthesis (HDR-NVS) remains challenged by geometric artifacts and radiometric distortions under multi-exposure conditions, primarily due to existing methods ignoring exposure and over-relying on color cues. Inspired by the integrated processing of color and structure of the human visual system (HVS), we propose Expo-GS, a novel framework that decomposes HDR-NVS into three interpretable components, namely, Irradiance Field Training, Geometry Field Training, and Interactive Joint Training. Central to Expo-GS is the exposure-aware signed distance function (Expo-SDF), which dynamically reweights geometric supervision via localized exposure reliability estimation, suppressing noisy gradients from unstable regions while enhancing structure learning in well-exposed areas. Building on this, we design an interactive optimization strategy that synchronizes Gaussian primitive growth and pruning with evolving Expo-SDF cues, enabling exposure-aware density control and eliminating hallucinated structures near exposure transitions. Experiments show that Expo-GS significantly outperforms prior methods on both synthetic and real-world datasets. It achieves a peak PSNR of 39.06 dB under HDR settings and up to 41.38 dB in the LDR-OE configuration, excelling in preserving high-frequency textures and maintaining structural consistency.

## 1 INTRODUCTION

Novel view synthesis (NVS) aims to generate photorealistic renderings from sparse input views Dalal et al. (2024), enabling continuous viewpoint interpolation for applications in virtual reality Xu et al. (2023), autonomous systems Hess et al. (2025), and creation of three-dimensional content Tang et al. (2023). However, conventional 8-bit low dynamic range (LDR) imaging struggles to capture the full radiometric complexity of real-world scenes, especially under extreme exposure, leading to degraded perceptual and geometric fidelity. In contrast, high dynamic range (HDR) imaging employs substantially higher per-channel bit depth to more faithfully capture the physical radiance, thereby mitigating the artifacts introduced by conventional nonlinear compression Chen et al. (2025).

Existing high dynamic range novel view synthesis (HDR-NVS) approaches can be broadly divided into two categories: NeRF-based methods Mildenhall et al. (2021); Martin-Brualla et al. (2021); Huang et al. (2022)) and 3D Gaussian Splatting (3D-GS) techniques ( Kerbl et al. (2023); Cai et al. (2024)). NeRF-based models encode scene radiance as continuous volumetric fields, enabling the synthesis of intricate lighting phenomena. Despite their expressiveness, these methods often fail to recover fine structures in shadowed or underexposed regions and suffer from heavy computational demands due to dense ray sampling. In contrast, 3D-GS offers higher rendering efficiency and has recently been adapted for NVS Fei et al. (2024). However, its design remains fundamentally tailored to LDR scenarios. Without explicit mechanisms to account for extreme exposure variations tend to generate hallucinated geometry and ghosting artifacts under challenging illumination conditions.

Current HDR-NVS approaches predominantly emphasize irradiance field modeling Cai et al. (2024), often relying on color regression as the primary strategy to approximate scene appearance under varying exposure conditions Huang et al. (2022). However, this single-modal learning paradigm poses two critical limitations: **(i)** Optimization driven solely by color cues fails to capture geometric

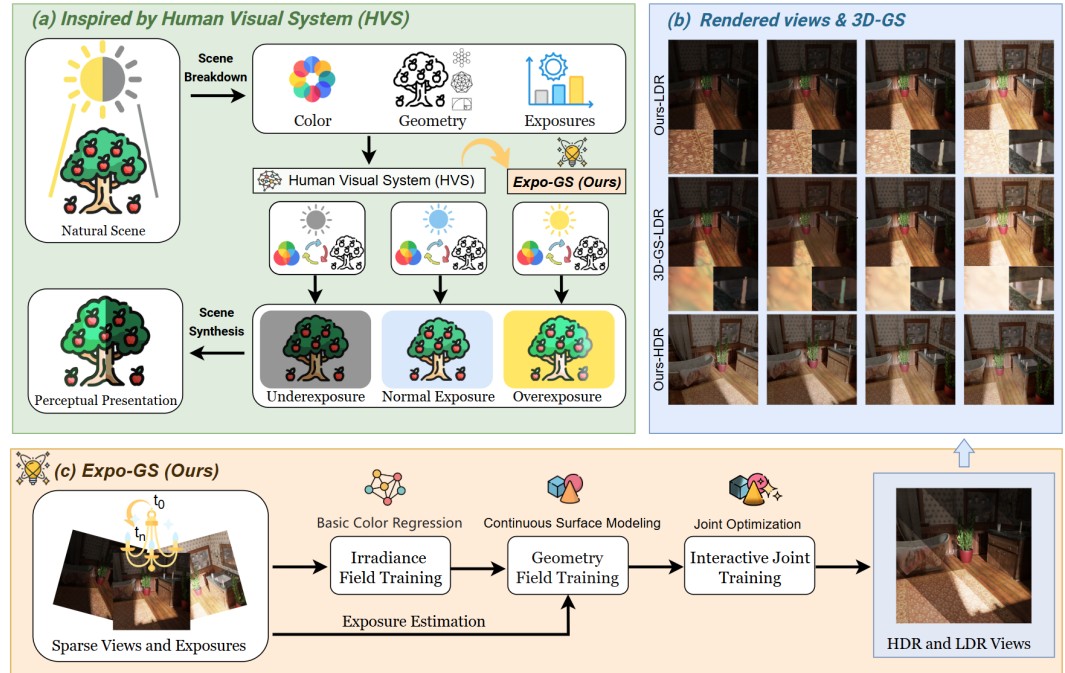

Figure 1: (a) Inspired by the human visual system (HVS), our method (Expo-GS) decomposes the scene into color, geometry, and exposure components to enable robust perception under varying exposure conditions. (b) Visualization comparisons demonstrate that Expo-GS substantially outperforms the baseline 3D-GS Kerbl et al. (2023) method in both LDR and HDR views. (c) Expo-GS devises a three-stage pipeline comprising irradiance field, geometry field, and interactive joint training.

variations induced by exposure disparities; and **(ii)** Regions with sharp luminance transitions, such as boundaries between illuminated and shadowed areas, are particularly vulnerable to radiance field distortions due to insufficient structural priors. Inspired by the human visual system (HVS), which functionally decouples color and geometry but integrates them for coherent perception, we introduce a biologically inspired framework (Figure 1) that jointly models irradiance and geometric structure.

In this paper, we present a novel framework, *Exposure-Aware Signed Distance Function in Gaussian Splatting (Expo-GS)*, which draws inspiration from the functional organization of the HVS. To address the limitations of existing HDR-NVS approaches, we introduce a disentangled-then-joint training paradigm that adaptively modulates radiometric and geometric supervision based on localized exposure conditions. Expo-GS seamlessly integrates color sensitivity and structural awareness into the GS pipeline, enabling accurate HDR scene reconstruction and robust performance in NVS. Unlike conventional SDF-based methods, our exposure-aware variant dynamically reweights geometric constraints according to the exposure reliability of observed regions. This adaptive mechanism mitigates overfitting in saturated or underexposed areas by down-weighting unreliable radiometric cues, while reinforcing supervision in well-exposed, structurally consistent regions, thus promoting stable and precise geometry estimation under extreme lighting conditions. To model radiance distribution, we adopt a soft forward cumulative rendering strategy during irradiance field training, facilitating efficient Gaussian projection and consistent scene visualization. Moreover, we introduce a joint optimization module to simultaneously refine color fidelity and geometric coherence, fostering a dynamic balance between radiometric precision and structural integrity. Together, these innovations enable Expo-GS to achieve high-fidelity reconstruction and generalization in complex HDR environments.

Our main contributions are summarized as follows:

- We propose a novel framework **Expo-GS** that, for the first time, conceptually decomposes HDR-NVS into three disentangled components: color, geometry, and exposure. This factorization improves HDR scene interpretability and supports precise, modular optimization.

- We introduce an exposure-aware signed distance function (**Expo-SDF**) that adaptively integrates geometric supervision across multi-exposure inputs. This mechanism enables reliable structural learning and significantly enhances geometric fidelity and consistency.
- We design a joint optimization strategy that couples Expo-SDF–guided geometry with refined irradiance reconstruction, enabling mutual optimization and producing sharper, more coherent results through a balanced integration of radiance fidelity and geometric stability.

## 2 RELATED WORK

**3D Gaussian Splatting (3D-GS).** 3D-GS has garnered significant attention for its real-time Qu et al. (2024); Wang et al. (2024); Hyun & Heo (2024); Wu et al. (2024b); Peng et al. (2024), differentiable rendering capabilities Feng et al. (2025); Lee et al. (2024); Yao et al. (2024). Its rapid development has been fueled by hierarchical Gaussian pruning and cross-domain extensions Yu et al. (2024); Chen & Lee (2024); Zhu et al. (2024). However, despite these advances, 3D-GS exhibits inherent limitations when applied to HDR-NVS Kerbl et al. (2023); Cai et al. (2024). Specifically, its reliance on spherical harmonic coefficients limits its ability to represent regions with extreme luminance variation Jiang et al. (2024); Liang et al. (2024). Furthermore, the absence of exposure-aware geometric supervision, due to inadequate regularization or uniform loss weighting across brightness levels, often results in structural artifacts and the loss of fine geometric details. These limitations underscore the necessity of jointly preserving radiometric accuracy and geometric consistency to achieve robust HDR-NVS.

**Signed Distance Function (SDF).** Neural networks can represent 3D geometry using continuous SDFs Chou et al. (2023); Park et al. (2019); Sitzmann et al. (2020); Choi et al. (2024), where surfaces are implicitly defined as the zero-level set of a neural field Zhang et al. (2024a); Yu et al. (2024); Choi et al. (2024). Integrating SDFs with volumetric rendering (e.g., VolSDF Yariv et al. (2021), NeuS Wang et al. (2021)) or 3D-GS-based methods (e.g., SuGaR Guédon & Lepetit (2024), GSDF Yu et al. (2024)) has demonstrated improved surface sharpness and structural coherence by combining precise geometric modeling with radiance-based robustness Zakharov et al. (2020); Mu et al. (2021); Chen et al. (2024); Cao & Taketomi (2024). However, existing SDF frameworks are inherently exposure-agnostic, as they typically assume fixed-exposure LDR inputs and fail to account for the exposure variations. Consequently, they often struggle to model overexposed or underexposed regions accurately, resulting in hallucinated geometry and blurring artifacts in NVS. This reveals a critical gap: the lack of exposure-aware SDF modeling capable of adapting to the challenges of HDR-NVS. For additional related work, please refer to the appendix C.

## 3 METHOD

### 3.1 IRRADIANCE FIELD TRAINING FOR BASIC COLOR REGRESSION

We adopt anisotropic Gaussians as the core representational primitives for HDR scene modeling, serving as fundamental units that enable efficient radiance field and geometric field representation:

$$\mathcal{G} = \{g_i = (\boldsymbol{\mu}_i, \boldsymbol{\Sigma}_i, \boldsymbol{\alpha}_i, k_i, \mathcal{M}_\theta)\} \tag{1}$$

where $\boldsymbol{\mu}_i$ is the Gaussian center, $\boldsymbol{\Sigma}_i$ the anisotropic covariance, $\boldsymbol{\alpha}_i$ the radiance coefficients, $k_i$ the scale factor, and $\mathcal{M}_\theta$ the shared SH parameters for radiance modeling. This formulation enables sub-pixel precision and high-frequency encoding under HDR conditions, while remaining compatible with LDR representations. To capture view-dependent HDR radiance, each Gaussian is assigned compact SH coefficients, yielding the observed radiance $\mathbf{c}_i^{\mathrm{HDR}} \in \mathbb{R}^3$ from direction $\mathbf{v} \in \mathbb{S}^2$:

$$\mathbf{c}_i^{\mathrm{HDR}}(\mathbf{v}) = \exp\left(\mathbf{B}(\mathbf{v})^\top \mathbf{k}_i\right) \tag{2}$$

Here, $\mathbf{B}(\mathbf{v}) \in \mathbb{R}^{(L+1)^2}$ denotes the real spherical harmonics (SH) basis evaluated at direction $\mathbf{v}$, and $\mathbf{k}_i \in \mathbb{R}^{(L+1)^2 \times 3}$ are the SH coefficients for the $i$-th point. To convert HDR radiance into LDR images, we employ a learnable tone mapping network that models the non-linear camera response under varying exposures. Instead of using a fixed function, we define tone mapping as a learnable transformation $\mathcal{M}_\theta : \mathbb{R}^3 \to \mathbb{R}^3$ operating in the log domain:

$$\mathbf{c}_i^{\mathrm{LDR}}(\Delta t) = \mathcal{M}_\theta\left(\log \mathbf{c}_i^{\mathrm{HDR}} + \log \Delta t\right) \tag{3}$$

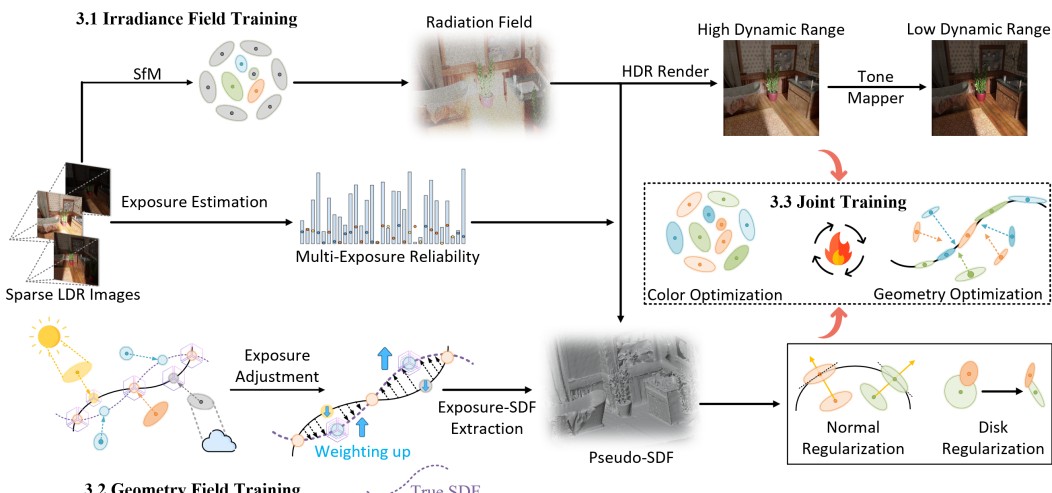

Figure 2: Expo-GS consists of three interpretable components—irradiance field training for color regression, geometry field training for exposure-aware structure modeling, and interactive joint training for refining both radiometric and geometric representations.

Here, $\Delta t$ denotes the exposure time and the logarithmic formulation stabilizes the dynamic range to facilitate learning. The network $\mathcal{M}_\theta$ is shared across all Gaussians and channels. We adopt a parallel, fully differentiable rasterization pipeline to project 3D Gaussians onto the 2D image plane. This simulates volumetric accumulation of semitransparent Gaussians. For a pixel $p$ at exposure time $\Delta t$, its color is computed using front-to-back soft blending of all overlapping Gaussians:

$$I(p \mid \Delta t) = \sum_{j \in \mathcal{N}_p} \mathbf{c}_j(\Delta t) \cdot \sigma_j \cdot \prod_{k=1}^{j-1}(1 - \sigma_k) \qquad (4)$$

In this expression, $\mathcal{N}_p$ denotes the depth-sorted set of Gaussians visible at pixel $p$. The term $\mathbf{c}_j(\Delta t) \in \mathbb{R}^3$ is the RGB radiance of the $j$-th Gaussian, modulated by exposure time via the learned tone mapping function. The visibility weight $\sigma_j \in [0, 1]$ encodes both opacity and projected spatial density. To model soft visibility and volumetric contribution, each Gaussian's support in screen space is represented by a continuous kernel. The visibility weight $\sigma_j$ at pixel $p$ is defined as:

$$\sigma_j = \alpha_j \cdot \exp\left(-\frac{1}{2}(p - \boldsymbol{\mu}_j)^\top \boldsymbol{\Sigma}_j^{-1}(p - \boldsymbol{\mu}_j)\right) \qquad (5)$$

where $\alpha_j$ is a learnable opacity scalar determining the base contribution of the Gaussian, and the exponential term models a 2D anisotropic Gaussian kernel centered at $\boldsymbol{\mu}_j$, shaped by $\boldsymbol{\Sigma}_j$. This formulation supports smooth, differentiable visibility estimation and compactly simulates soft occlusion during rasterization. We adopt a unified photometric loss that supports both HDR and tone-mapped LDR supervision, enabling flexible optimization across radiometric and perceptual domains:

$$\mathcal{L}_{\text{render}} = \sum_{j=1}^{B}\left[\mathcal{L}_1(\mathcal{R}_j, \mathcal{R}_j^{\text{gt}}) + \lambda \cdot \mathcal{L}_{\text{D-SSIM}}(\mathcal{R}_j, \mathcal{R}_j^{\text{gt}})\right], \qquad (6)$$

where $\mathcal{R}_j$ is the rendered output at the $j$-th viewpoint, and $\mathcal{R}_j^{\text{gt}}$ is the corresponding ground truth, either in HDR or tone-mapped LDR format depending on the training stage. This formulation is robust to exposure variations and promotes alignment in both pixel-wise and perceptual spaces.

### 3.2 Geometry Field Training for Exposure-Aware Structural Modeling

To better model continuous geometric structures in HDR scenes, we propose an exposure-aware signed distance field (Expo-SDF) informed by Gaussian visibility. Prior work has shown that explicit

geometric supervision improves irradiance field optimization. Building on this, we introduce an Expo-SDF-based density formulation that attenuates contributions from overexposed or underexposed regions, while emphasizing well-exposed areas to enhance geometric fidelity. To integrate multi-exposure reliability into the geometry field, we first define an exposure estimation function:

$$\mathcal{E}_i(q) = e_i \cdot \max_{c \in \{R,G,B\}} I_{i,c}(q), \tag{7}$$

where $q$ denotes a pixel coordinate in the LDR image $I_i$, $e_i$ is the exposure factor (exposure time) for image $i$, and $I_{i,c}(q)$ is the intensity of channel $c$. This formulation provides a simple yet effective proxy for assessing the radiometric reliability of individual pixels. Additional design analysis is provided in appendix D. We then project each Gaussian center $\boldsymbol{\mu}_j$ into image $i$ to retrieve its exposure estimate as $\mathcal{E}(\boldsymbol{\mu}_j) = \mathcal{E}_i(\pi_i(\boldsymbol{\mu}_j))$, where $\pi_i(\cdot)$ denotes the projection function. Using these estimates, we define an exposure-normalized density function that down-weights unreliable contributions:

$$\dot{d}(p) = \sum_j \frac{\alpha_j}{\mathcal{E}(\boldsymbol{\mu}_j) + \epsilon} \cdot \exp\left(-\frac{1}{2}(p - \boldsymbol{\mu}_j)^\top \boldsymbol{\Sigma}_j^{-1}(p - \boldsymbol{\mu}_j)\right), \tag{8}$$

where $\dot{d}(p)$ denotes the exposure-normalized Gaussian density at 3D location $p$, $\alpha_j$ is the learnable opacity scalar of the $j$-th Gaussian, $\boldsymbol{\mu}_j \in \mathbb{R}^3$ is its 3D center, $\boldsymbol{\Sigma}_j \in \mathbb{R}^{3 \times 3}$ is the anisotropic covariance matrix encoding its spatial extent and orientation, $\mathcal{E}(\boldsymbol{\mu}_j)$ is the projected exposure value as defined in Eq. (7), and $\epsilon$ is a small constant to avoid division by zero. This formulation effectively suppresses the influence of Gaussians located in radiometrically unreliable regions, while preserving the contributions from well-exposed observations. The derivation and proof are provided in appendix B. To estimate surface proximity in regions lacking reliable exposure or dense geometry supervision, we construct a pseudo-Expo-SDF based on the aggregated Gaussian density:

$$f_{\text{HDR}}(p) = \pm s_{g^*} \cdot \sqrt{-2 \log \dot{d}(p)}, \tag{9}$$

where $g^*$ denotes the index of a locally dominant Gaussian around $p$ under the standard front-to-back compositing, and $s_{g^*}$ is a local scale factor derived from its spatial extent. Both $s_{g^*}$ and the associated normal direction are only used to provide a local scale and orientation for interpreting the smooth potential $-\log \dot{d}(p)$ as a distance-like quantity; they do not modify the underlying log-density field. This formulation provides a smooth and differentiable approximation of the underlying geometry, even in regions with unreliable exposure, and serves as a practical surrogate for mesh initialization, SDF alignment, and supervision pruning. To ensure geometric consistency between the estimated pseudo-Expo-SDF and the ground-truth surface, we introduce an exposure-aware supervision loss:

$$\mathcal{L}_{\text{SDF}}^{\text{HDR}} = \frac{1}{|P|} \sum_{p \in P} \left| \hat{f}(p) - f_{\text{HDR}}(p) \right|, \tag{10}$$

where $\hat{f}(p)$ denotes the reference signed distance value at point $p$, obtained via mesh-based rendering or camera-projected depth, and $f_{\text{HDR}}(p)$ is the pseudo-Expo-SDF derived from exposure-aware Gaussian density. This loss enables soft geometric supervision in radiometrically challenging areas, leveraging structural priors without requiring complete or noise-free ground-truth annotations. To further guide geometry learning in radiometrically ambiguous regions, we regularize the pseudo-Expo-SDF field with two complementary constraints: normal alignment and spatial flatness. Specifically, we encourage the gradient of the Expo-SDF field at each point $p \in P$ to align with the dominant surface normal $\mathbf{n}_{g^*}$, promoting local planar consistency:

$$\mathcal{L}_{\text{normal}} = \frac{1}{|P|} \sum_{p \in P} \left(1 - \left\langle \frac{\nabla f_{\text{HDR}}(p)}{\|\nabla f_{\text{HDR}}(p)\|}, \mathbf{n}_{g^*} \right\rangle\right)^2. \tag{11}$$

Moreover, to ensure that each Gaussian approximates a locally planar surface element, we introduce a disk regularization term that promotes disk-like anisotropic configurations. Rather than directly minimizing the smallest principal axis—which may result in unstable gradients—we adopt a differentiable softmin-based formulation to enable smooth and effective optimization:

$$\mathcal{L}_{\text{disk}} = \frac{1}{|\mathcal{G}|} \sum_{g \in \mathcal{G}} -\tau \cdot \log\left(\sum_{i=1}^{3} e^{-s_i^g / \tau}\right), \tag{12}$$

where $s_i^g$ denotes the scale of Gaussian $g$ along its $i$-th principal axis, and $\tau$ is a temperature parameter that controls the sharpness of the softmin approximation. This formulation encourages each Gaussian to adopt a geometrically coherent, anisotropic structure that aligns with the local surface geometry, thereby improving training stability and spatial regularity.

## 3.3 INTERACTIVE JOINT TRAINING FOR HARMONIZING OPTIMIZATION

In this stage, the irradiance and geometry fields are jointly refined via a coupled optimization strategy, which is essential for high-fidelity HDR-NVS. Accurate geometric structures guide light propagation near object boundaries and shadows, while radiometric cues help recover fine-grained geometry. Central to this process is a geometry-aware density control mechanism that dynamically adjusts the spatial distribution of Gaussians based on feedback from the exposure-aware signed distance field $f_{\text{HDR}}$. The growth activation for a candidate Gaussian center $\mathbf{c}$ is defined as:

$$\epsilon_g = \nabla_g + \omega_s \cdot \exp\left(-\frac{f_{\text{HDR}}(\mathbf{c})^2}{2\sigma^2}\right) + \omega_n \cdot \left(1 - \|\nabla f_{\text{HDR}}(\mathbf{c})\|\right), \tag{13}$$

where $\nabla_g$ is the accumulated training gradient, and the remaining terms encourage growth near the zero-level surface while suppressing updates in uncertain regions. Gaussians with $\epsilon_g > \tau_g$ are duplicated with perturbations to refine confident structures. Conversely, pruning is guided by a reliability-based score:

$$\epsilon_p = \sigma_a - \omega_p \cdot \left(1 - \exp\left(-\frac{f_{\text{HDR}}(\mathbf{c})^2}{2\sigma^2}\right)\right), \tag{14}$$

where $\sigma_a$ is the accumulated opacity. Gaussians with $\epsilon_p < \tau_p$ are removed from further optimization, reducing clutter and suppressing artifacts. By integrating growth and pruning, the geometry-aware density control aligns the Gaussian layout with the evolving Expo-SDF, reinforcing consistency between appearance and structure. To enable stable joint training under such dynamic updates, we combine radiance and geometry losses into a unified objective:

$$\mathcal{L}_{\text{joint}} = \mathcal{L}_{\text{render}} + \lambda_{\text{SDF}} \cdot \mathcal{L}_{\text{SDF}}^{\text{HDR}} + \lambda_{\text{normal}} \cdot \mathcal{L}_{\text{normal}} + \lambda_{\text{disk}} \cdot \mathcal{L}_{\text{disk}}, \tag{15}$$

where $\mathcal{L}_{\text{render}}$ supervises radiance reconstruction, and the remaining terms ensure geometric consistency under varying exposures. The Interactive Joint Training strategy enables geometric structures to inform radiance field modeling, while radiometric cues, in turn, refine fine-grained geometric details. This bidirectional guidance significantly enhances both visual fidelity and structural consistency.

## 4 EXPERIMENTS

### 4.1 EXPERIMENTAL SETTINGS

**Dataset.** We conduct our experiments using HDR-NeRF Huang et al. (2022), a dataset designed for NVS under HDR conditions. This dataset features multi-view, multi-exposure captures across 8 synthetic scenes and 4 real-world scenes. Each scene is recorded from 35 distinct viewpoints, with five different exposure levels per view. Each synthetic scene is accompanied by a corresponding HDR reference stored in the OpenEXR (.exr) format. Following prior work Cai et al. (2024); Wu et al. (2024c), we use HDR images from 18 viewpoints for training, each paired with one randomly sampled LDR exposure from $\{t_1, t_3, t_5\}$. The remaining 17 viewpoints are reserved for evaluation.

**Experimental Details.** Our framework is trained in three stages. The initialization of Gaussian point parameters follows the settings in 3D-GS Kerbl et al. (2023), while the tone mapping network is adopted from HDR-GS Cai et al. (2024). We first perform 8,000 iterations of 3D-GS-based warm-up to establish an initial discrete radiance field. This is followed by 12,000 iterations of implicit training on the Expo-SDF to refine the geometry and extract a high-fidelity surface point cloud. Finally, we conduct 10,000 iterations of joint optimization to balance radiance accuracy and geometric consistency. The entire training pipeline is implemented in PyTorch and optimized with Adam Zhang et al. (2024b), running for a total of 30,000 iterations on a single NVIDIA RTX A6000 GPU.

| Method | LDR-OE $(t_1, t_3, t_5)$ | | | LDR-NE $(t_2, t_4)$ | | | HDR | | |
|---|---|---|---|---|---|---|---|---|---|
| | PSNR↑ | SSIM↑ | LPIPS↓ | PSNR↑ | SSIM↑ | LPIPS↓ | PSNR↑ | SSIM↑ | LPIPS↓ |
| NeRF Mildenhall et al. (2021) | 15.63 | 0.612 | 0.309 | 17.83 | 0.652 | 0.353 | – | – | – |
| 3D-GS Kerbl et al. (2023) | 22.37 | 0.690 | 0.276 | 18.97 | 0.778 | 0.309 | – | – | – |
| NeRF-W Martin-Brualla et al. (2021) | 29.83 | 0.936 | 0.047 | 29.22 | 0.927 | 0.050 | – | – | – |
| HDR-NeRF Huang et al. (2022) | 39.35 | 0.979 | 0.028 | 36.58 | 0.954 | 0.028 | 36.12 | 0.908 | 0.019 |
| HDR-GS Cai et al. (2024) | 41.06 | 0.980 | 0.013 | 36.36 | 0.973 | 0.019 | 37.98 | 0.976 | 0.014 |
| **Expo-GS (Ours)** | **41.38** | **0.989** | **0.010** | **37.47** | **0.984** | **0.014** | **39.06** | **0.981** | **0.010** |

Table 1: Quantitative comparison on synthetic datasets. LDR-OE and LDR-NE represent LDR-NVS settings using exposure subsets $\{t_1, t_3, t_5\}$ and $\{t_2, t_4\}$, respectively. HDR represents NVS based on complete HDR reconstruction. Our method achieves the best performance across all configurations.

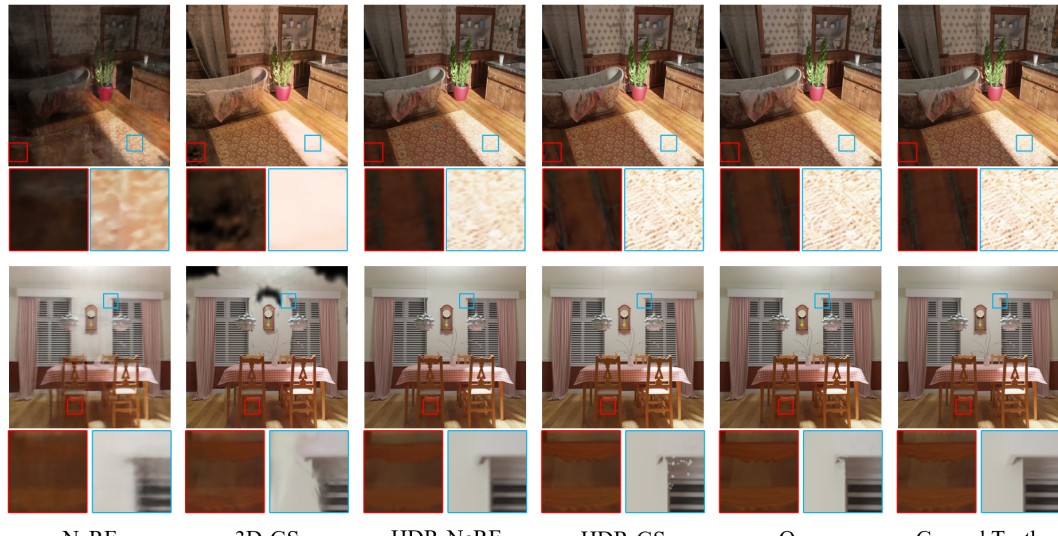

| NeRF | 3D-GS | HDR-NeRF | HDR-GS | Ours | Ground Truth |

Figure 3: Qualitative comparisons on synthetic datasets. Our method achieves superior color fidelity and geometric accuracy over baseline 3D-GS, with zoom-in regions revealing detailed improvements.

## 4.2 RESULTS

**Quantitative Comparisons on Synthetic Datasets.** Table 1 provides a comprehensive quantitative assessment of NVS performance across synthetic datasets under three distinct exposure configurations: LDR with exposure ($\{t_1, t_3, t_5\}$), LDR with exposure ($\{t_2, t_4\}$), and HDR. Our proposed methodology, Expo-GS, demonstrates superior performance compared to existing approaches across virtually all evaluation metrics with the sole exception being PSNR under the LDR-NE configuration. Furthermore, within the HDR domain, Expo-GS exhibits exceptional efficacy, registering a PSNR of 39.06 dB, which exceeds HDR-NeRF Huang et al. (2022) by 2.94 dB and HDR-GS Cai et al. (2024) by 1.08 dB. Our method achieves superior perceptual fidelity, evidenced by the lowest LPIPS and highest SSIM, demonstrating its ability to preserve visual authenticity and geometric consistency.

**Qualitative Comparisons on Synthetic Datasets.** Figure 3 presents qualitative comparisons across multiple approaches on synthetic scenes. Traditional NeRF Mildenhall et al. (2021) and 3D-GS Kerbl et al. (2023) suffer from severe color smearing and geometric distortions, particularly in regions with challenging illumination. While HDR-NeRF Huang et al. (2022) and HDR-GS Cai et al. (2024) enhance visual realism to some extent, they still exhibit noticeable color inconsistencies and geometric blurring under extreme lighting transitions. In contrast, our Expo-GS method reconstructs both radiance and geometry with high fidelity. Fine-grained textures—such as wood grain, fabric patterns, and shadow boundaries—are clearly preserved and visually aligned with the ground truth.

**Quantitative Comparisons on Real-World Datasets.** As shown in Table 2, our Expo-GS framework achieves consistently superior performance across all exposure conditions. Under the LDR-OE setting, it attains a PSNR of 35.59 dB, SSIM of 0.981, and LPIPS of 0.020, significantly outperforming HDR-GS Cai et al. (2024) and HDR-NeRF Huang et al. (2022). Even in the more challenging LDR-NE scenario, it maintains robust performance with the highest PSNR of 32.17 dB and the lowest LPIPS of 0.033. The results demonstrate robust generalization and resilience to exposure changes.

| Method | LDR-OE $(t_1, t_3, t_5)$ | | | LDR-NE $(t_2, t_4)$ | | |
|---|---|---|---|---|---|---|
| | PSNR↑ | SSIM↑ | LPIPS↓ | PSNR↑ | SSIM↑ | LPIPS↓ |
| NeRF Mildenhall et al. (2021) | 16.21 | 0.682 | 0.295 | 15.84 | 0.716 | 0.227 |
| 3D-GS Kerbl et al. (2023) | 18.32 | 0.849 | 0.113 | 20.38 | 0.719 | 0.161 |
| NeRF-W Martin-Brualla et al. (2021) | 29.39 | 0.914 | 0.087 | 29.17 | 0.918 | 0.086 |
| HDR-NeRF Huang et al. (2022) | 32.35 | 0.939 | 0.065 | 32.71 | 0.947 | 0.071 |
| HDR-GS Cai et al. (2024) | 34.94 | 0.962 | 0.031 | 31.24 | 0.953 | 0.045 |
| **Expo-GS (Ours)** | **35.59** | **0.981** | **0.020** | **32.17** | **0.972** | **0.033** |

Table 2: Quantitative comparisons on real-world datasets. Our method attains markedly superior overall performance, demonstrating its practical effectiveness and feasibility in real-world scenarios.

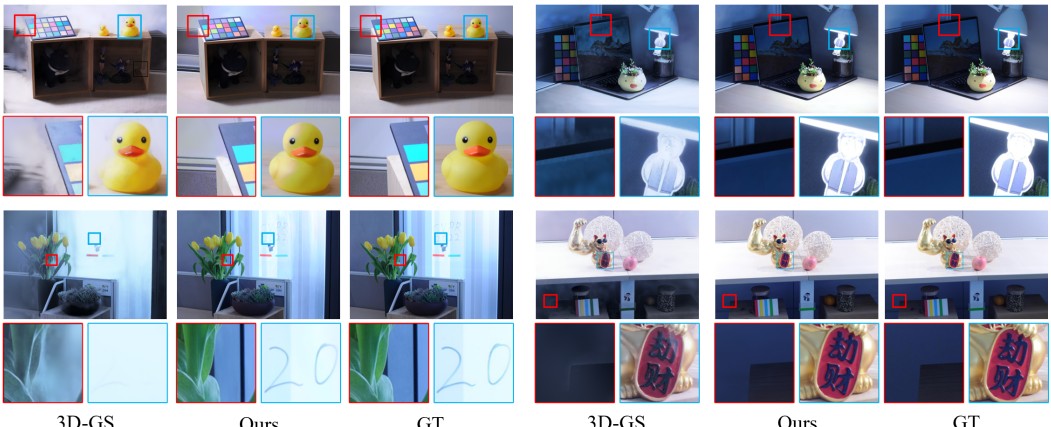

3D-GS      Ours      GT      3D-GS      Ours      GT

Figure 4: Qualitative comparisons on real-world datasets. Our method achieves superior color fidelity and geometric accuracy over other methods, with zoom-in regions revealing detailed improvements.

**Qualitative Comparisons on Real-World Datasets.** Figure 4 further demonstrates qualitative improvements over 3D-GS Kerbl et al. (2023). In overexposed regions (e.g., light bulbs, duck), our method suppresses saturation and geometric distortion in overexposed regions and recovers fine detail in underexposed areas; exposure-guided supervision further enhances high-frequency reconstruction, collectively reducing radiometric bias and structural artifacts under challenging real-world settings.

| CRF Domain | Linear | Logarithmic | Reinhard | ACES |
|---|---|---|---|---|
| HDR | 29.57 | 38.77 | 38.91 | 38.82 |
| LDR-OE | 33.49 | 41.33 | 40.14 | 40.65 |
| LDR-NE | 32.82 | 37.18 | 36.82 | 36.64 |

| Exposure | $\{t_3\}$ | $\{t_1, t_5\}$ | $\{t_1, t_3, t_5\}$ | $\{t_1, \ldots, t_5\}$ |
|---|---|---|---|---|
| HDR | 24.17 | 32.81 | 38.64 | 38.98 |
| LDR-OE | 23.78 | 35.44 | 41.66 | 41.69 |
| LDR-NE | 23.13 | 34.25 | 36.81 | 37.53 |

Table 3: Ablation study on distinct CRF methods and exposure time sets on synthetic datasets.

| Method | LDR-OE $(t_1, t_3, t_5)$ | | | LDR-NE $(t_2, t_4)$ | | | HDR | | |
|---|---|---|---|---|---|---|---|---|---|
| | PSNR↑ | SSIM↑ | LPIPS↓ | PSNR↑ | SSIM↑ | LPIPS↓ | PSNR↑ | SSIM↑ | LPIPS↓ |
| Baseline | 22.37 | 0.690 | 0.276 | 18.97 | 0.778 | 0.309 | — | — | — |
| Irradiance (8K) | 29.17 | 0.912 | 0.055 | 29.86 | 0.914 | 0.063 | 28.86 | 0.894 | 0.091 |
| Geometry (12K) | 31.92 | 0.929 | 0.049 | 31.78 | 0.923 | 0.057 | 30.29 | 0.915 | 0.086 |
| Geometry + 1K | 36.41 | 0.961 | 0.031 | 35.39 | 0.955 | 0.026 | 34.76 | 0.963 | 0.037 |
| Joint (10K) | 41.38 | 0.989 | 0.010 | 37.47 | 0.984 | 0.014 | 39.06 | 0.981 | 0.010 |

Table 4: Ablation study on training stages highlighting the importance of geometry field training.

**Ablation study.** In Table 3, the results demonstrate that nonlinear CRFs consistently outperform the linear variant and the Logarithmic transform achieves the highest PSNR scores (41.33 and 37.18 at LDR setting, respectively), suggesting its effectiveness in enhancing LDR image quality through contrast adjustment. And we evaluate the impact of different exposure time combinations. The results reveal model performance improves progressively with increased diversity in the exposure set. In

Table 4, We observed that once the pseudo-SDF is established, the initial iterations of the third stage (Geometry+1k training) exhibit a noticeable improvement in performance. This suggests that the quality of SDF learning indirectly influences the photometric fidelity and geometric consistency of the synthesized views. It means second stage plays an essential intermediary role in the framework.

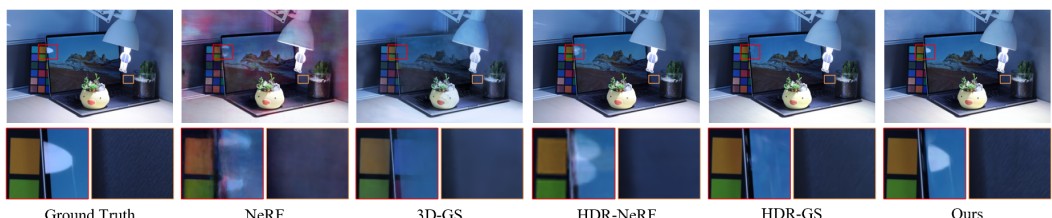

Figure 5: Geometric comparison of different methods in real-world scenes with specular reflections.

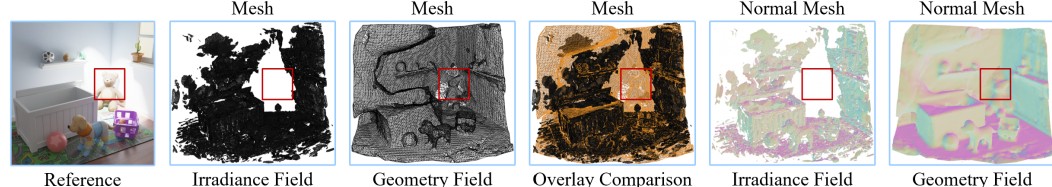

Figure 6: Geometric comparisons of training after first and second stage in synthetic scenarios.

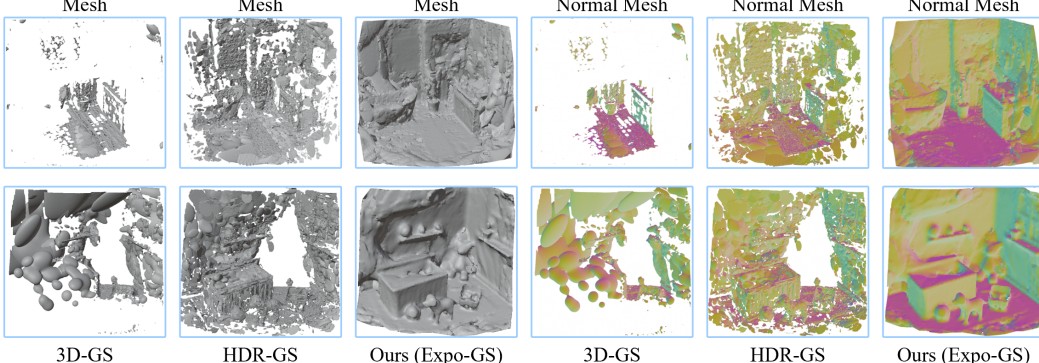

Figure 7: Geometric comparisons of different methods in synthetic scenarios using mesh visualization.

**Geometric comparisons.** Figure 5 showcases a challenging scenario with pronounced specularities and high-frequency texture. Our Expo-GS framework reconstructs both mirror-like surfaces and adjacent diffuse regions with high fidelity, preserving subtle detail and faithfully modeling non-Lambertian behavior. Figure 6 isolates the contribution of the geometry training stage, showing marked improvements in structural modeling and contour delineation; this stage further refines the point cloud distribution. Figure 7 extends the comparison to prior methods: 3D-GS produces coarse geometry with noise and artifacts, while HDR-GS, despite improved irradiance fidelity, exhibits unstable geometry. In contrast, our approach achieves higher geometric accuracy, smoother surface continuity, and superior normal consistency, thereby distinguishing itself from existing HDR-NVS techniques. Taken together, these results indicate that our method advances irradiance reconstruction while simultaneously delivering robust geometric representation. Additional results in appendix F.

## 5 CONCLUSION

We present Expo-GS, a biologically inspired HDR-NVS framework that jointly models radiance and geometry under varying exposures. Through Expo-SDF and a decoupled-then-joint training paradigm, our method alleviates radiometric bias and geometric inconsistency caused by illumination extremes. To our knowledge, this is the first framework to unify multi-exposure radiometric and geometric modeling in HDR-NVS, offering novel insights into exposure-aware scene reconstruction.

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

APPENDIX

# A LIMITATIONS

Although Expo-GS performs well on HDR-NVS, it also has limitations. Our joint optimization introduces additional training overhead because of the multi-stage schedule and the Expo-SDF refinement. The moderate increase in training time mainly comes from the second stage (Geometry Field Training). However, when combined with the single-pass rendering strategy in the first stage and the lightweight optimization in the third stage, the overall training time remains reasonable, and our method still outperforms other methods in terms of inference speed. As shown by the main HDR-NVS experiments, our method achieves a favorable balance between performance and efficiency, where "(8 / 21 / 7)" denotes the relative training cost of the three stages. Additionally, the reliance on multi-exposure inputs requires strictly static scenes, which limits applicability to dynamic settings. Future work will explore extensions to time-varying scenes in real-world environments.

# B THEORETICAL DERIVATIONS

## B.1 ANALYSIS OF EQUATION (8)

Equation (8) defines the *exposure-normalized Gaussian density*:

$$\dot{d}(\mathbf{p}) = \sum_j \frac{\alpha_j}{E(\boldsymbol{\mu}_j) + \varepsilon} \exp\left[-\tfrac{1}{2}(\mathbf{p} - \boldsymbol{\mu}_j)^\top \boldsymbol{\Sigma}_j^{-1}(\mathbf{p} - \boldsymbol{\mu}_j)\right], \tag{8}$$

where each Gaussian's opacity coefficient $\alpha_j$ is scaled by the estimated exposure $E(\boldsymbol{\mu}_j)$ at its center, with a small constant $\varepsilon$ added to ensure numerical stability. This exposure-aware attenuation preserves the influence of well-exposed regions while down-weighting overexposed highlights and severely underexposed, noise-prone areas.

The resulting density field $\dot{d}(\mathbf{p})$ is strictly positive, infinitely differentiable, and globally bounded. Moreover, it is monotonically *decreasing* with increasing exposure, enabling robust structural modeling in HDR conditions. Notably, the scaled negative logarithm of $\dot{d}(\mathbf{p})$ induces a quadratic form in $\mathbf{p}$, which serves as the foundation for the pseudo-signed distance function defined in Equation (9) and utilized in geometric supervision.

## B.2 PROOF OF EQUATION (8)

1. **Positivity and smoothness.**
   Each term in the summation is positive: $\alpha_j > 0$, $E(\boldsymbol{\mu}_j) + \varepsilon > 0$, and the Gaussian exponential kernel is positive and $C^\infty$. Therefore, $\dot{d}(\mathbf{p})$ is strictly positive and infinitely differentiable over $\mathbb{R}^3$.

2. **Global upper bound.**
   Since the denominator is lower-bounded by $\varepsilon$, we have:
   $$\dot{d}(\mathbf{p}) \leq \sum_j \frac{\alpha_j}{\varepsilon},$$
   ensuring $\dot{d}(\mathbf{p})$ is globally bounded and $\log \dot{d}(\mathbf{p})$ is finite.

3. **Monotonicity with respect to exposure.**
   Differentiating each term w.r.t. $E(\boldsymbol{\mu}_j)$:
   $$\frac{\partial}{\partial E(\boldsymbol{\mu}_j)}\left(\frac{\alpha_j}{E(\boldsymbol{\mu}_j) + \varepsilon} e^{-\frac{1}{2}(\cdots)}\right) = -\frac{\alpha_j}{(E(\boldsymbol{\mu}_j) + \varepsilon)^2} e^{-\frac{1}{2}(\cdots)} < 0,$$
   indicating $\dot{d}(\mathbf{p})$ decreases monotonically with local exposure.

4. **Local dominance.**
   When a single Gaussian $g_k$ dominates locally:
   $$\dot{d}(\mathbf{p}) \approx \frac{\alpha_k}{E_k + \varepsilon} \exp\left[-\tfrac{1}{2}(\mathbf{p} - \boldsymbol{\mu}_k)^\top \boldsymbol{\Sigma}_k^{-1}(\mathbf{p} - \boldsymbol{\mu}_k)\right], \quad E_k := E(\boldsymbol{\mu}_k).$$

5. **Quadratic structure of the log-density.**
   Taking the negative logarithm:

   $$-2\log \dot{d}(\mathbf{p}) = (\mathbf{p} - \boldsymbol{\mu}_k)^\top \boldsymbol{\Sigma}_k^{-1}(\mathbf{p} - \boldsymbol{\mu}_k) + C,$$

   where $C = -2\log \alpha_k + 2\log(E_k + \varepsilon)$ is constant in $\mathbf{p}$.

6. **Euclidean upper bound.**
   Let $\boldsymbol{\Sigma}_k = \mathrm{diag}(s_{k1}^2, s_{k2}^2, s_{k3}^2)$, and define $s_{\min} := \min_i s_{ki}$. Then:

   $$-2\log \dot{d}(\mathbf{p}) \geq \frac{\|\mathbf{p} - \boldsymbol{\mu}_k\|_2^2}{s_{\min}^2}.$$

7. **Gradient alignment.**
   The gradient is:

   $$\nabla_{\mathbf{p}}\dot{d} = \dot{d}\,\boldsymbol{\Sigma}_k^{-1}(\boldsymbol{\mu}_k - \mathbf{p}),$$

   which aligns with the Gaussian's outward normal, enabling normal-aligned supervision.

8. **Conclusion.**
   Equation (8) defines a well-behaved, exposure-aware surrogate density. Its negative logarithm induces a Mahalanobis distance, forming the mathematical basis for the pseudo-SDF used in Equation (9).

## B.3 ANALYSIS OF EQUATION (9)

Equation (9) defines a *pseudo-signed distance function (pseudo-SDF)*:

$$f_{\mathrm{HDR}}(\mathbf{p}) = \pm s_{g^*}\sqrt{-2\log \dot{d}(\mathbf{p})}, \tag{9}$$

where $g^*$ denotes a locally dominant Gaussian near $p$, and $s_{g^*}$ is a local scale parameter derived from its principal axes (e.g., lower-bounded by its smallest axis length). This formulation transforms the log-density into a distance-like scalar field whose sign is determined by the orientation of the local surface normal. Importantly, $g^*$ and $s_{g^*}$ are introduced here as analytical devices for interpreting the smooth potential $-\log \dot{d}(p)$; in our implementation, $f_{\mathrm{HDR}}(p)$ is evaluated directly from $\dot{d}(p)$ without differentiating through any hard nearest-Gaussian or min-axis selection. Consequently, $f_{\mathrm{HDR}}(\mathbf{p})$ behaves like a true SDF in well-exposed regions and provides reliable geometric cues in radiometrically ambiguous areas.

## B.4 PROOF OF EQUATION (9)

1. **Dominant Gaussian assumption.**
   Assume $g^*$ dominates in the local neighborhood:

   $$\dot{d}(\mathbf{p}) \approx \frac{\alpha_{g^*}}{E(\boldsymbol{\mu}_{g^*}) + \varepsilon}\exp\left[-\tfrac{1}{2}(\mathbf{p} - \boldsymbol{\mu}_{g^*})^\top \boldsymbol{\Sigma}_{g^*}^{-1}(\mathbf{p} - \boldsymbol{\mu}_{g^*})\right].$$

2. **Log transformation.**
   Taking $-2\log$:

   $$-2\log \dot{d}(\mathbf{p}) = (\mathbf{p} - \boldsymbol{\mu}_{g^*})^\top \boldsymbol{\Sigma}_{g^*}^{-1}(\mathbf{p} - \boldsymbol{\mu}_{g^*}) + C.$$

3. **Principal axis decomposition.**
   Diagonalizing $\boldsymbol{\Sigma}_{g^*}$:

   $$(\mathbf{p} - \boldsymbol{\mu}_{g^*})^\top \boldsymbol{\Sigma}_{g^*}^{-1}(\mathbf{p} - \boldsymbol{\mu}_{g^*}) = \sum_{i=1}^{3} \frac{\Delta_i^2}{s_{g^*,i}^2}.$$

4. **Distance upper bound.**
   Let $s_{g^*} = \min_i s_{g^*,i}$. Then:

   $$-2\log \dot{d}(\mathbf{p}) \geq \frac{\|\mathbf{p} - \boldsymbol{\mu}_{g^*}\|_2^2}{s_{g^*}^2}.$$

5. **Distance approximation.**
   Rescaling gives:

   $$s_{g^*}\sqrt{-2\log \dot{d}(\mathbf{p})} \approx \|\mathbf{p} - \boldsymbol{\mu}_{g^*}\|_2.$$

6. **Sign function.**
   Using the local surface normal $\mathbf{n}_{g^*}$:

   $$\mathrm{sign}(f_{\mathrm{HDR}}(\mathbf{p})) = \mathrm{sign}\left((\mathbf{p} - \boldsymbol{\mu}_{g^*}) \cdot \mathbf{n}_{g^*}\right).$$

7. **Definition of pseudo-SDF.**
   Combining the above, we define:

   $$f_{\mathrm{HDR}}(\mathbf{p}) = \pm s_{g^*}\sqrt{-2\log \dot{d}(\mathbf{p})}.$$

8. **Gradient alignment.**
   Given

   $$\nabla_{\mathbf{p}}\dot{d} = \dot{d}\,\boldsymbol{\Sigma}_{g^*}^{-1}(\boldsymbol{\mu}_{g^*} - \mathbf{p}),$$

   the gradient $\nabla f_{\mathrm{HDR}}$ is aligned with the local normal $\mathbf{n}_{g^*}$.

9. **Conclusion.**
   The function $f_{\mathrm{HDR}}(\mathbf{p})$ behaves as a smooth, exposure-aware approximation of a signed distance field, suitable for geometry regularization under HDR settings.

Whether a region is up-weighting or down-weighting therefore depends jointly on $\alpha_j$ and $\mathcal{E}(\mu_j)$, rather than on $\mathcal{E}(\mu_j)$ alone. In overexposed regions, $\mathcal{E}(\mu_j)$ becomes large, so it appears in the denominator and strongly attenuates the contribution of the corresponding Gaussians. In underexposed regions, however, $\alpha_j$ becomes very small: in regions with extreme underexposure, the low-signal conditions lead to unstable gradients, and the learned Gaussian opacities tend to be close to zero. As a result, the ratio $\dfrac{\alpha_j}{\mathcal{E}(\mu_j) + \epsilon}$ becomes very small (with $\epsilon > 0$ fixed), so the contribution of underexposed Gaussians is also effectively suppressed.

## C  EXTENDED RELATED WORK

**High Dynamic Range (HDR) Imaging.** Traditional HDR imaging hinges on multi-exposure fusion to reconstruct the full luminance spectrum of a scene Reinhard (2020); Kang et al. (2003). Early methods focused on radiometric calibration to counteract nonlinear camera responses Zhang et al. (2025); Liu et al. (2025); Huang et al. (2024); Jun-Seong et al. (2022); Wu et al. (2024a), while recent works integrate deep learning with physical sensor models, using attention-based fusion and physics-inspired tone mapper for enhanced fidelity Seetzen et al. (2023); Zhang & Yau (2009). HDR-NeRF Huang et al. (2022) introduces HDR capabilities into Neural Radiance Fields via inverse gamma correction and HDR-aware training, improving reconstructions in low-light regions. HDR-GS Cai et al. (2024) further advances this by adapting 3D-GSKerbl et al. (2023) to HDR synthesis, leveraging dual dynamic range point cloud modeling and differentiable rasterization for simultaneous HDR-LDR reconstruction with exposure control. However, most HDR methods emphasize pixel-wise radiometric accuracy and rely heavily on color regression, overlooking the intricate coupling between lighting variations and geometry—especially problematic in 3D scenes where structural consistency is crucial.

**Human Visual System (HVS).** HVS possesses a remarkable ability to perceive both color and structural information of scenes under highly dynamic and varying lighting conditions Thorpe et al. (1996). This capacity arises from several key perceptual mechanisms Parraga et al. (2000); Banks et al. (2012).

First, exposure adaptation allows the human eye to function across a wide luminance range—from dim starlight at night to bright sunlight at noon—through dynamic gain control by photoreceptor cells in the retina Goodale & Haffenden (1998). Second, contrast sensitivity drives the visual system to focus on local edges and changes in luminance rather than absolute pixel intensity Adini et al. (2002). This mechanism is essential for capturing object boundaries and geometric structure. Third, the HVS performs dual-stream processing to separately process color and structure Field et al. (1993).

Specifically, cone cells in the retina transmit fine-grained chromatic and texture information through the parvocellular pathway (color stream) to the visual cortex, while rod cells are more sensitive to motion and coarse structure, channeling information through the magnocellular pathway (structure stream) Goebel et al. (2004); Salin & Bullier (1995). These two streams are later recombined in higher-level visual areas to construct a coherent and stable perception of the visual world Toosy et al. (2004); Lu & Sperling (2001).

Inspired by the HVS, a more robust approach is to decouple color (i.e., irradiance) and structure (i.e., geometry) during the training process. Specifically, radiometric and geometric attributes of the scene are modeled separately and then jointly optimized at the perceptual level. This mimics the human strategy of treating different exposure regions with varying sensitivity and integrating information based on local reliability. Thus, the perceptual principles of the HVS provide a biologically grounded motivation for decoupled modeling in HDR-NVS. Emulating the HVS paradigm of "first decomposing, then integrating" enables the development of HDR-NVS systems that are not only photometrically faithful but also geometrically stable across varying lighting conditions.

# D  In-Depth Design Analysis

In multi-exposure or extreme dynamic range scenarios, geometry optimized solely from pixel color residuals often fails. First, gradients from saturated or underexposed pixels are either zero or unstable, causing the density $\sigma$ to expand indiscriminately and leading to surface drift. Second, exposure times ($\Delta t$) vary across views; after the camera's nonlinear response function, residual magnitudes become imbalanced, further compounding depth inconsistencies.

These issues are well-documented in prior work: *RawNeRF* demonstrates that NeRF fails to recover highlights and shadows under LDR conditions, while *Gaussian-DK* reports severe ghosting artifacts in *3D-GS* under high-contrast lighting. In summary, ignoring exposure reliability and relying solely on color cues hinders robust geometric reconstruction under varying exposure conditions—an observation supported by both physical modeling and empirical evidence.

Although the exposure time $\Delta t$ is a global setting, the local exposure conditions within a single frame are far from uniform; bright highlights and deep shadows often coexist in our dataset. As early as the HDR fusion method of Debevec&Malik (1997), the pixel weight $w(z)$ was given a bell-shaped profile to down-weight overexposed and underexposed samples. The spirit is identical here: we assess pixel-wise confidence from exposure, rather than just blindly using the global $\Delta t$.

In the bathroom scene of the synthetic dataset, the red flowerpot provides a representative test case. Under normal exposure ($t_3$), the average RGB values are approximately $P_a \approx (161, 74, 58)$, with the red channel at 161. Upon inspection, we observed that these pixels retain full weight—and in some cases even receive slightly increased weight—during subsequent computation and training. Under overexposure ($t_5$), the same region exhibits average RGB values of $P_b \approx (222, 170, 144)$, with the red channel rising to 222. In this case, the SDF weight assigned to $P_b$ is slightly reduced, which is appropriate given that local highlights introduce noisy color residuals that can destabilize pseudo-SDF supervision.

If a large, uniform area has a high $\max(\mathrm{RGB})$ value, it is beneficial to reduce the weight of its contribution to the SDF. Consider an area that is almost pure red $(255, 0, 0)$ and contains little geometric or textural detail. Assigning the sparsely distributed 3D point cloud to this area is sufficient. Reducing its density does not affect the appearance, and the boundary geometry is still defined by the surrounding well-exposed pixels. Therefore, the weighting scheme does not misclassify vibrant colors; on the contrary, it highlights their geometric value and contributes to more stable optimization. And our goal is not to calculate the precise absolute brightness of pixels but to quickly determine whether a pixel is within a reliable (neither overexposed/underexposed) range for subsequent SDF geometric gradient weighting.

# E  Hyperparameter Configuration for Expo-GS Training

We detail the hyperparameter configuration used across the three stages of Expo-GS: irradiance field training, geometry field training, and interactive joint training.

| Parameter | Value | Description |
|---|---|---|
| **1. General Training Settings** | | |
| Optimizer | Adam | Used in all training stages |
| Learning Rate (Stage 1) | 2.5e-3 | Irradiance field training |
| Learning Rate (Stage 2) | 1.0e-3 | Geometry field training |
| Learning Rate (Stage 3) | 5.0e-4 | Joint optimization |
| Total Iterations | 30,000 | 8000 (Stage 1) + 12000 (Stage 2) + 10000 (Stage 3) |
| Hardware | NVIDIA RTX A6000 | 49 GB VRAM (49140 MiB), Single-GPU |
| Framework | PyTorch | CUDA 11.8 compatible |
| | Imambi et al. (2021) | Paszke et al. (2019) |
| **2. Geometry Field Training Settings** | | |
| Expo-SDF Loss | $\lambda_{\text{SDF}} = 0.2$ | Geometry field stage |
| Normal Alignment Loss | $\lambda_{\text{normal}} = 0.2$ | Geometry field stage |
| Disk Regularization Loss | $\lambda_{\text{disk}} = 0.1$ | Geometry field stage |
| **3. Joint Optimization Settings** | | |
| Expo-SDF Loss | $\lambda_{\text{SDF}} = 0.1$ | Joint stage |
| Normal Alignment Loss | $\lambda_{\text{normal}} = 0.1$ | Joint stage |
| Disk Regularization Loss | $\lambda_{\text{disk}} = 0.05$ | Joint stage |

Table 5: Hyperparameter configuration across training stages for Expo-GS, comprising Stage 1 (irradiance field training), Stage 2 (geometry field training), and Stage 3 (interactive joint training).

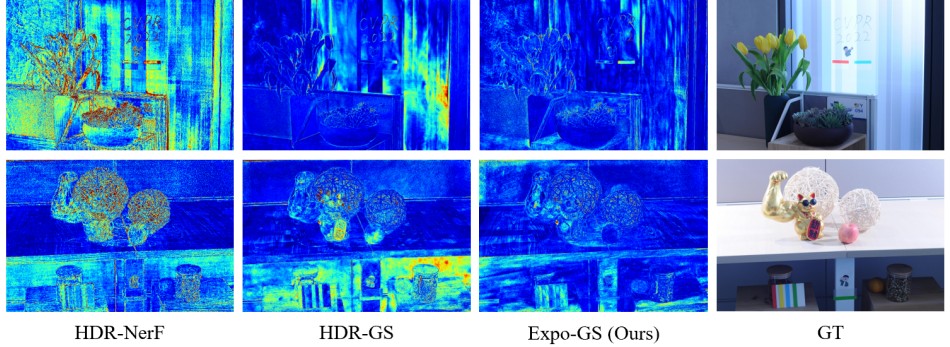

HDR-NerF    HDR-GS    Expo-GS (Ours)    GT

Figure 8: Comparison of error maps with state-of-the-art methods.

## F  EXTENDED RESULTS

**Tone mapper.** The core limitation of fixed tone mappers lies in their generalizability. Their parameters are empirical, and when faced with cross-scene or extreme exposure conditions, manual re-tuning is often required to avoid color bias and detail loss. In contrast, learnable tone mappers can adaptively map relationships end-to-end without human intervention. Furthermore, nearly all previous HDR-NVS methods (HDR-NeRF, HDR-GS, etc.) inherently rely on learnable tone mappers. If we were to manually select the optimal fixed tone mappers for each of them, it would introduce new unfair factors. Therefore, we retain the learnable tone mapper to maintain end-to-end properties and ensure that comparisons with existing methods focus on exposure modeling capabilities rather than the tone mapper itself.

**Evaluation.** To evaluate the performance of HDR-NVS, we adopt three widely used metrics: Peak Signal-to-Noise Ratio (PSNR) Johnson (2006), Structural Similarity Index Measure (SSIM) Wang (2004), and Learned Perceptual Image Patch Similarity (LPIPS) Zhang et al. (2018). PSNR Johnson (2006) directly quantifies radiometric fidelity through pixel-level differences, while SSIM Wang (2004) and LPIPS Zhang et al. (2018) indirectly reflect geometric coherence by measuring structural and perceptual similarity, respectively. Together, these metrics provide a dual-perspective assessment that captures both irradiance accuracy and geometric consistency.

| Method | LDR-OE ($t_1, t_3, t_5$) | | | LDR-NE ($t_2, t_4$) | | | HDR | | |
|---|---|---|---|---|---|---|---|---|---|
| | PSNR↑ | SSIM↑ | LPIPS↓ | PSNR↑ | SSIM↑ | LPIPS↓ | PSNR↑ | SSIM↑ | LPIPS↓ |
| GaussHDR Liu et al. (2025) | 39.74 | 0.974 | 0.021 | 36.89 | 0.977 | 0.018 | 37.66 | 0.967 | 0.025 |
| Mono-HDR Zhang et al. (2025) | 30.16 | 0.923 | 0.051 | 31.44 | 0.938 | 0.055 | 38.81 | 0.975 | 0.014 |
| LTM-NeRF Huang et al. (2024) | 39.42 | 0.976 | 0.033 | 36.67 | 0.964 | 0.027 | 36.55 | 0.953 | 0.018 |
| HDR-Plenoxels Jun-Seong et al. (2022) | 36.38 | 0.959 | 0.045 | 35.02 | 0.943 | 0.045 | 33.86 | 0.941 | 0.046 |
| **Expo-GS (Ours)** | **41.38** | **0.989** | **0.010** | **37.47** | **0.984** | **0.014** | **39.06** | **0.981** | **0.010** |

Table 6: More comparisons with additional methods.

**More comparisons with additional methods.** All comparison methods are based on the training and testing set partitions declared in Section 4.1 of our paper. Experimental results 6 demonstrate that our method still achieves a significant lead, outperforming additional methods in both overall reconstruction quality and perceptual consistency across LDR-OE, LDR-NE, and HDR settings.

| Method | LDR-OE ($t_1, t_3, t_5$) | | | LDR-NE ($t_2, t_4$) | | | HDR | | |
|---|---|---|---|---|---|---|---|---|---|
| | PSNR↑ | SSIM↑ | LPIPS↓ | PSNR↑ | SSIM↑ | LPIPS↓ | PSNR↑ | SSIM↑ | LPIPS↓ |
| Expo-GS (w/ Vanilla SDF) | 36.59 | 0.954 | 0.038 | 34.36 | 0.948 | 0.044 | 32.95 | 0.947 | 0.046 |
| **Expo-GS (w/ Expo-SDF)** | **41.38** | **0.989** | **0.010** | **37.47** | **0.984** | **0.014** | **39.06** | **0.981** | **0.010** |

Table 7: Comparisons with Vanilla SDF.

**Comparisons with Vanilla SDF.** The experiments 7 demonstrated that the vanilla SDF fails to deliver satisfactory results and cannot reasonably handle the exposure conditions of HDR-NVS. We attribute this to the fact that the vanilla SDF forced constraints fail to properly process exposure information, leading to field geometry conflicts. This causes conflicts between irradiance field and geometry field especially in the overexposed and underexposed regions. This experiment further demonstrates the effectiveness of Expo-SDF.

| Method | LDR-OE ($t_1, t_3, t_5$) | | | LDR-NE ($t_2, t_4$) | | | HDR | | |
|---|---|---|---|---|---|---|---|---|---|
| | PSNR↑ | SSIM↑ | LPIPS↓ | PSNR↑ | SSIM↑ | LPIPS↓ | PSNR↑ | SSIM↑ | LPIPS↓ |
| Standard SDF Guédon & Lepetit (2024) | 31.52 | 0.928 | 0.061 | 31.27 | 0.924 | 0.065 | 30.71 | 0.922 | 0.079 |
| SuGaR Guédon & Lepetit (2024) | 36.10 | 0.954 | 0.049 | 35.28 | 0.958 | 0.049 | 33.36 | 0.943 | 0.048 |
| GSDF Yu et al. (2024) | 38.76 | 0.966 | 0.026 | 36.39 | 0.971 | 0.022 | 34.81 | 0.957 | 0.033 |
| PulledGS Zhang et al. (2024a) | 39.69 | 0.971 | 0.019 | 36.07 | 0.965 | 0.027 | 35.95 | 0.962 | 0.025 |
| **Expo-GS (Ours)** | **41.38** | **0.989** | **0.010** | **37.47** | **0.984** | **0.014** | **39.06** | **0.981** | **0.010** |

Table 8: Comparisons with SDF-based methods reveal their limitations under LDR inputs.

**Comparisons with SDF-based methods.** As shown in Table 8, existing SDF-based methods (e.g., SuGaR Guédon & Lepetit (2024), GSDF Yu et al. (2024), and PulledGS Zhang et al. (2024a)), though effective under fixed-exposure LDR conditions, exhibit suboptimal performance in HDR-NVS scenarios. These methods typically employ dense surface guiding while uniformly weighting multi-exposure inputs, neglecting the exposure disparities inherent in LDR observations. Consequently, they fail to handle the challenging conditions inherent in HDR configurations, leading to the emergence of spurious geometries and degraded radiometric fidelity. Notably, our proposed Expo-GS framework effectively accommodates multi-exposure inputs by incorporating exposure-aware optimization, achieving superior results across all evaluated metrics.

Table 9 reveals that under the LDR-OE setting (exposure levels observed during training), both NeRFactor and ReLight-NeRF deliver reasonable performance owing to their architectural illumination modeling and the availability of ground-truth supervision. The learned mappings between brightness, geometry, and material remain effective within the trained exposure range, yielding relatively high-quality reconstructions. However, under the more challenging LDR-NE setting (unseen exposures), both methods suffer a substantial degradation in performance. This drop is attributed to the absence of explicit exposure modeling, which compels the networks to implicitly approximate brightness differences across views. Consequently, unseen exposure times induce shifts in activation distributions, leading to poor generalization and a sharp decline in image quality.

While NeRFactor and ReLight-NeRF are effective for modeling illumination variations within LDR conditions, their outputs are ultimately governed by apparent brightness. As such, they remain competent for LDR relighting tasks but are unsuitable for HDR-NVS or cross-exposure generalization.

| Method | LDR-OE ($t_1, t_3, t_5$) | | | LDR-NE ($t_2, t_4$) | | | HDR | | |
|---|---|---|---|---|---|---|---|---|---|
| | PSNR↑ | SSIM↑ | LPIPS↓ | PSNR↑ | SSIM↑ | LPIPS↓ | PSNR↑ | SSIM↑ | LPIPS↓ |
| NeRFactor Zhang et al. (2021) | 33.16 | 0.941 | 0.038 | 18.62 | 0.761 | 0.325 | – | – | – |
| ReLight-NeRF Toschi et al. (2023) | 36.58 | 0.957 | 0.029 | 21.31 | 0.819 | 0.292 | – | – | – |
| Expo-GS(Ours) | 41.38 | 0.989 | 0.010 | 37.47 | 0.984 | 0.014 | 39.06 | 0.981 | 0.010 |

Table 9: Comparisons with relightable methods reveal their limitations under LDR inputs.

| Exposure | $t_3$ | $\{t_7, t_5\}$ | $\{t_6, t_5\}$ | $\{t_1, t_5\}$ | $\{t_1, t_3, t_5\}$ | $\{t_1, \ldots, t_5\}$ |
|---|---|---|---|---|---|---|
| HDR | 24.17 | 32.14 | 33.58 | 32.81 | 38.64 | 38.98 |
| LDR-OE | 23.78 | 34.93 | 35.37 | 35.44 | 41.66 | 41.69 |
| LDR-NE | 23.13 | 33.61 | 33.98 | 34.25 | 36.81 | 37.53 |

Table 10: Generalization of exposure distribution.

**Generalization of exposure distribution.** To evaluate the model under more extreme exposure conditions, we first use the learned tone mapper to synthesize LDR images at two additional extreme exposure times, $t_6 = 1/16\,\text{s}$ and $t_7 = 1/32\,\text{s}$. The framework reconstructs an HDR representation of the scene, scales this HDR signal by the desired exposure factor, and then passes it through the learned tone mapper to obtain an LDR image corresponding to an exposure level never seen during training (see Eq. 3 in the main paper). We then conduct experiments using these synthetically generated LDR images under extreme exposure conditions, and summarize the results on the synthetic dataset in terms of PSNR in Table 10.

When the input comprises only two extreme exposure levels, $\{1/32\,\text{s}, 32\,\text{s}\}$ (i.e., $\{t_7, t_5\}$), the model still achieves a PSNR of 32–35 dB across all three evaluation settings (HDR, LDR-OE, and LDR-NE), which is generally regarded as visually acceptable. As the exposure interval decreases, the PSNR increases steadily, indicating that reconstruction quality improves as the exposure gap narrows. Nevertheless, even the most extreme two-exposure configuration yields acceptable results ($> 30\,\text{dB}$). These experiments demonstrate the robustness of our method across a broad range of exposure conditions and confirm its ability to handle exposure variations even in extreme scenarios.

**Validating the significance of the Geometry Field Stage.**

From Table 4, we observe that once the pseudo-SDF has been established, the initial iterations of the third stage (Geometry+1k training) yield a noticeable improvement in performance metrics. This suggests that the quality of SDF learning indirectly influences the photometric fidelity and geometric consistency of the synthesized views. As a geometric prior, it plays an essential intermediary role in the overall framework, underscoring the significance of the geometry field training stage.

In fact, the three-stage training is a common GS pipeline. In our experiments, a 12k-iteration geometry field stage serves as a stable configuration for all scenarios, including complex synthetic and real-world environments. We found that convergence is usually achievable without 12k. Table 11 summarizes the results of training our Geometry Field Stage for 3k, 6k, 9k, and 12k iterations.

As shown in our additional ablation on the number of iterations in the Geometry Field stage, increasing the geometry training from 3K to 9K iterations leads to substantial improvements, while the performance saturates between 9K and 12K. This demonstrates that (i) the geometry field stage is indeed crucial for converging to high-quality HDR geometry, and (ii) the method is not overly sensitive to the exact iteration count once a sufficient number of iterations ($\approx 9K$) is used.

**Efficiency Comparison.** As shown in Table 12, the moderate increase in training time is due to the second stage (Geometry Field Training). However, combined with the single rendering strategy of the first stage and the lightweight optimization of the third stage, the overall training time is not very long, and our method outperforms other methods in terms of inference speed. In fact, based on the main experimental results of HDR-NVS, our method achieves a balance between performance and efficiency.

| Method | LDR-OE $(t_1, t_3, t_5)$ | | | LDR-NE $(t_2, t_4)$ | | | HDR | | |
|---|---|---|---|---|---|---|---|---|---|
| | PSNR↑ | SSIM↑ | LPIPS↓ | PSNR↑ | SSIM↑ | LPIPS↓ | PSNR↑ | SSIM↑ | LPIPS↓ |
| Geometry Field/ 3K | 36.83 | 0.956 | 0.496 | 35.66 | 0.948 | 0.043 | 34.37 | 0.944 | 0.043 |
| Geometry Field/ 6K | 38.32 | 0.965 | 0.412 | 36.18 | 0.960 | 0.035 | 35.88 | 0.948 | 0.028 |
| Geometry Field/ 9K | **41.41** | 0.982 | **0.010** | 37.43 | 0.982 | **0.014** | 39.01 | 0.977 | 0.011 |
| **Geometry Field/ 12K** | 41.38 | **0.989** | **0.010** | **37.47** | **0.984** | **0.014** | **39.06** | **0.981** | **0.010** |

Table 11: Sensitivity analysis of geometry field training.

| Method | Training time (min) | GPU memory (GB) | Inference speed (fps) |
|---|---|---|---|
| HDR-NeRF | 517 | 80 | 0.128 |
| HDR-GS | 33 | 8 | 122 |
| **Expo-GS (ours)** | 36 (8 / 21 / 7) | 6 / 11 / 9 | 131 |

Table 12: Training efficiency and inference speed comparison.

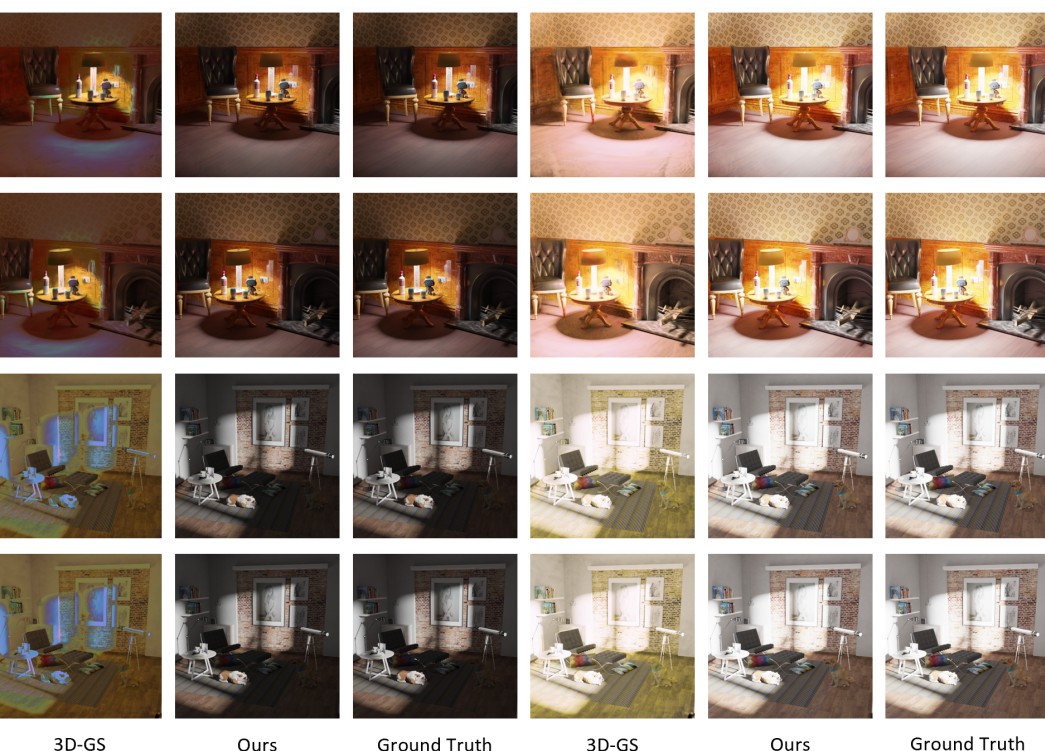

| 3D-GS | Ours | Ground Truth | 3D-GS | Ours | Ground Truth |

Figure 9: Qualitative comparisons on synthetic datasets. Our method achieves superior color fidelity and geometric accuracy over baseline 3D-GS under LDR settings.

**Qualitative Comparison.** Figure 9 illustrates qualitative results on representative synthetic scenes under challenging illumination. The baseline 3D-GS method exhibits prominent artifacts, such as color bleeding, oversaturation, and structure deformation, especially around high-contrast regions like candles, windows, and shadow boundaries. In contrast, our method (Expo-GS) demonstrates significantly improved radiometric accuracy and structural coherence. Reflective surfaces and high-frequency textures (e.g., wallpaper patterns, fabric details, window frames) are faithfully reconstructed, closely matching the ground truth. These improvements highlight the effectiveness of exposure-aware SDF supervision and joint optimization in mitigating hallucinations and ensuring geometric stability across varying exposure conditions.

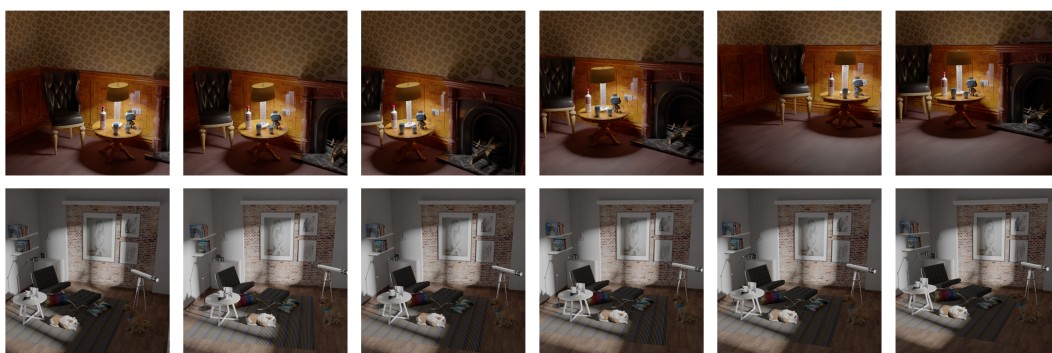

Figure 10: Our method achieves superior color fidelity and geometric accuracy under HDR settings.

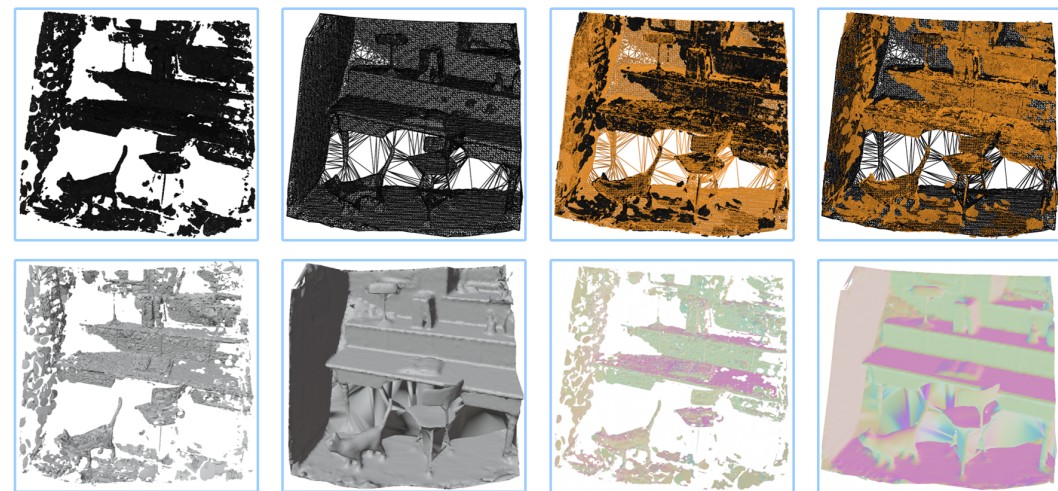

Figure 11: Geometric comparisons of training after first and second stage in synthetic scenarios.

**Geometric comparison.** As illustrated in the Figure 11, we provide qualitative visualizations of both the mesh and the corresponding normal maps obtained after the first-stage irradiance field training and the subsequent geometry field training. The irradiance field training alone is insufficient for reliable geometric modeling, as the initial triangular mesh derived from the point cloud displays prominent holes and structural discontinuities. By contrast, the geometry field training, enhanced through Expo-SDF optimization, produces a smoother and more coherent scene geometry. The normal visualizations further highlight the improved geometric fidelity, exhibiting strong surface consistency and continuity. Overall, these qualitative results demonstrate that geometry field training effectively captures structural information and reconstructs scene geometry, mitigating noise and stripe artifacts, filling missing regions, and substantially improving both surface smoothness and normal coherence.

**Ethics Statement.**

We adhere to ethics statement and have no content requiring disclosure.

**Reproducibility Sstatement.**

Table 5 provides detailed hyperparameter information, and the code will be open-sourced later.

**The Use of Large Language Models (LLMs).**

We have only used LLM for minor paragraph polishing.

