# OpenReview forum: "Expo-GS: Exposure-Aware Signed Distance Function in Gaussian Splatting for High Dynamic Range"
_ICLR.cc/2026/Conference — Submitted to ICLR 2026_

### Official Review · Reviewer_JLWc · 2025-10-28

**Soundness:** 3
**Presentation:** 3
**Contribution:** 2
**Rating:** 4
**Confidence:** 3

**Summary:**

This paper proposes Expo-GS, an exposure-aware framework for HDR novel view synthesis (HDR-NVS). It introduces an Exposure-aware Signed Distance Function (Expo-SDF) integrated into 3D Gaussian Splatting (3D-GS), and a three-stage pipeline consisting of: 1. Irradiance field training for color regression; 2. Geometry field training via exposure-modulated SDF; and 3. Interactive joint optimization of geometry and radiance. The paper claims improved geometric fidelity and radiometric consistency under multi-exposure HDR conditions.

**Strengths:**

+ The work addresses HDR-NVS, a challenging setting not well covered in Gaussian Splatting literature.
+ Introducing exposure reliability weighting into geometric supervision is conceptually interesting.
+ Multiple ablation studies and qualitative comparisons demonstrate the framework’s impact on geometry and color reconstruction.

**Weaknesses:**

- The experiments are limited to the HDR-NeRF dataset, with training and testing conducted under similar exposure distributions. It remains unclear how well Expo-GS generalizes to unseen exposure levels, camera response curves, or dynamic lighting variations. It is beneficial to test the model on cross-dataset or real-world outdoor scenes where exposure and tone mapping differ significantly.
- The training strategy (8k + 12k + 10k iterations) seems complex. However, the paper provides little analysis of its sensitivity or convergence behavior. How crucial is the geometry field stage for the final results?
- The derivation of Expo-SDF assumes that a single Gaussian dominates locally, which may not hold in regions of dense overlap or transparency. How does the pseudo-SDF behave when multiple Gaussians contribute comparable densities?

**Questions:**

Please refer to the Weaknesses.

---

> ### Author Response · Authors · 2025-11-24
> **Response to Weakness 1 for Reviewer JLWc**
>
> We appreciate your thoughtful review and are grateful for the positive feedback on our focus on the challenging HDR-NVS setting, the conceptual novelty of our approach. For the weaknesses you identified, we provide further clarifications and additional experiments below.
>
> ### **Weakness1. Extend the generalizability of our method.**
>
>
> We have divided the extension of generalizability into three sub-directions: exposure distribution, camera response function (CRF), and dynamic lighting variations.
>
>
> ### **1.** Exposure Distribution Generalization
>
> Although the HDR-NVS benchmark provides only five LDR images per viewpoint in its synthetic training set—captured at t1 = 0.125 s, t2 = 0.5 s, t3 = 2 s, t4 = 8 s, and t5 = 32 s, we first exploit the learned tone-mapper to synthesize LDR images at two additional, extreme exposures, namely t6 = 1/16 s and t7 = 1/32s.
>
> Concretely, the framework reconstructs an HDR representation of the scene. This HDR signal is then scaled by the desired exposure factor and passed through the learned tone mapper, producing an LDR image corresponding to an exposure never encountered during training (see Eq. 3 in the main paper). Then we conducted experiments using LDR images under extreme exposure conditions.
>
>
> | **Exposure** | *t*₃ | {*t*₇, *t*₅} | {*t*₆, *t*₅} | {*t*₁, *t*₅} | {*t*₁, *t*₃, *t*₅} | {*t*₁, ..., *t*₅} |
> |:-------------|:----:|:-----------:|:-----------:|:-----------:|:-----------------:|:----------------:|
> | HDR | 24.17 | 32.14 | 33.58 | 32.81 | 38.64 | 38.98 |
> | LDR-OE | 23.78 | 34.93 | 35.37 | 35.44 | 41.66 | 41.69 |
> | LDR-NE | 23.13 | 33.61 | 33.98 | 34.25 | 36.81 | 37.53 |
>
>
> When the input comprises only two extreme exposure levels—{1/32s, 32 s} ({t7, t5})—the model still achieves a PSNR of 32-35 dB across all three evaluation settings (HDR / LDR-OE / LDR-NE), reaching a level generally considered visually acceptable. As the exposure interval decreases, PSNR increases steadily, indicating that reconstruction quality improves as the exposure gap narrows. Nevertheless, even the most extreme two-exposure configurations yield acceptable results (>30 dB).
>
>
> Interestingly, under dual extreme exposure settings—such as {t7, t5}, {t6, t5}—the model achieves PSNR values that are 7-10 dB higher than those obtained with the single normal exposure t3. This result clearly demonstrates that combining exposures across a wide dynamic range significantly improves the model’s ability to recover fine details in both highlight and shadow regions.
>
>
> These experiments demonstrate the robustness of our method across a broad range of exposure conditions, confirming its capacity to perceive exposure variations even under extreme scenarios.
>
>
> ### **2.** Camera Response Function (CRF) Generalization
>
>
> | **CRF Domain** | Linear | Logarithmic | Reinhard | ACES |
> |:---------------|:------:|:-----------:|:--------:|:----:|
> | HDR | 29.57 | 38.77 | 38.91 | 38.82 |
> | LDR-OE | 33.49 | 41.33 | 40.14 | 40.65 |
> | LDR-NE | 32.82 | 37.18 | 36.82 | 36.64 |
>
>
>
> We compare four commonly used Camera Response Function (CRF) transformations: Linear, Logarithmic, Reinhard and ACES (Academy Color Encoding System). The results demonstrate that nonlinear CRFs consistently outperform the linear variant and the Logarithmic transform achieves the highest PSNR scores (41.33 and 37.18 at LDR setting, respectively), suggesting its effectiveness in enhancing LDR image quality through contrast adjustment.
>
>
> ### **3.** Robustness to Dynamic Lighting Variations
>
>
>
> | Stage | | **LDR-OE** | | | **LDR-NE** | | | **HDR** | |
> |:------|:-----:|:-----:|:------:|:-----:|:-----:|:------:|:-----:|:-----:|:------:|
> | | PSNR↑ | SSIM↑ | LPIPS↓ | PSNR↑ | SSIM↑ | LPIPS↓ | PSNR↑ | SSIM↑ | LPIPS↓ |
> | NeRFactor | 33.16 | 0.941 | 0.038 | 18.62 | 0.761 | 0.325 | - | - | - |
> | ReLight-NeRF | 36.58 | 0.957 | 0.029 | 21.31 | 0.819 | 0.292 | - | - | - |
> | **Expo-GS (Ours)** | **41.38** | **0.989** | **0.010** | **37.47** | **0.984** | **0.014** | **39.06** | **0.981** | **0.010** |
>
>
> This table reveals that under the LDR-OE setting (exposure levels observed during training), both NeRFactor and ReLight-NeRF deliver reasonable performance owing to their architectural illumination modeling and the availability of ground-truth supervision. The learned mappings between brightness, geometry, and material remain effective within the trained exposure range, yielding relatively high-quality reconstructions. However, under the more challenging LDR-NE setting (unseen exposures), both methods suffer a substantial degradation in performance. This drop is attributed to the absence of explicit exposure modeling, which compels the networks to implicitly approximate brightness differences across views. Consequently, unseen exposure times induce shifts in activation distributions, leading to poor generalization and a sharp decline in image quality.
>
>
> Thank you for your suggestion. We have also added these experiments to the manuscript (see Tables 3, 9, and 10).

---

> ### Author Response · Authors · 2025-11-24
> **Response to Weakness 2 for Reviewer JLWc**
>
> ### **Weakness2. Validating the significance of the Geometry Field Stage.**
>
>
>
> | Method | | **LDR-OE** | | | **LDR-NE** | | | **HDR** | |
> |:-------|:-----:|:-----:|:------:|:-----:|:-----:|:------:|:-----:|:-----:|:------:|
> | | PSNR↑ | SSIM↑ | LPIPS↓ | PSNR↑ | SSIM↑ | LPIPS↓ | PSNR↑ | SSIM↑ | LPIPS↓ |
> | Baseline | 22.37 | 0.690 | 0.276 | 18.97 | 0.778 | 0.309 | — | — | — |
> | Irradiance (8K) | 29.17 | 0.912 | 0.055 | 29.86 | 0.914 | 0.063 | 28.86 | 0.894 | 0.091 |
> | Geometry (12K) | 31.92 | 0.929 | 0.049 | 31.78 | 0.923 | 0.057 | 30.29 | 0.915 | 0.086 |
> | Geometry + 1K | 36.41 | 0.961 | 0.031 | 35.39 | 0.955 | 0.026 | 34.76 | 0.963 | 0.037 |
> | Joint (10K) | 41.38 | 0.989 | 0.010 | 37.47 | 0.984 | 0.014 | 39.06 | 0.981 | 0.010 |
>
>
> Although the second stage (geometry field training) does not lead to significant improvements in direct image reconstruction metrics, its core function lies in guiding the model to establish a stable and structurally consistent pseudo-SDF surface. This geometric representation serves as a critical prior for the subsequent interactive joint training stage.
>
>
> We observed that once the pseudo-SDF is established, the initial iterations of the third stage (Geometry+1k training) exhibit a noticeable improvement in performance metrics. This suggests that the quality of SDF learning indirectly influences the photometric fidelity and geometric consistency of the synthesized views. As a geometric prior, it plays an essential intermediary role in the overall framework, underscoring the significance of the geometry field training stage.
>
>
> In fact, the three-stage training is a common GS pipeline. In our experiments, a 12k-iteration geometry field stage serves as a stable configuration for all scenarios, including complex synthetic and real-world environments. We found that convergence is usually achievable without 12k. The results obtained by training our Geometry Field Stage at 3k, 6k, 9k, and 12k are as follows:
>
>
> | Method | | **LDR-OE** | | | **LDR-NE** | | | **HDR** | |
> |:-------|:-----:|:-----:|:------:|:-----:|:-----:|:------:|:-----:|:-----:|:------:|
> | | PSNR↑ | SSIM↑ | LPIPS↓ | PSNR↑ | SSIM↑ | LPIPS↓ | PSNR↑ | SSIM↑ | LPIPS↓ |
> | Geometry Field/ 3K | 36.83 | 0.956 | 0.496 | 35.66 | 0.948 | 0.043 | 34.37 | 0.944 | 0.043 |
> | Geometry Field/ 6K | 38.32 | 0.965 | 0.412 | 36.18 | 0.960 | 0.035 | 35.88 | 0.948 | 0.028 |
> | Geometry Field/ 9K | 41.41 | 0.982 | 0.010 | 37.43 | 0.982 | 0.014 | 39.01 | 0.977 | 0.011 |
> | **Geometry Field/ 12K** | **41.38** | **0.989** | **0.010** | **37.47** | **0.984** | **0.014** | **39.06** | **0.981** | **0.010** |
>
> As shown in our additional ablation on the number of iterations in the Geometry Field stage, increasing the geometry training from 3K to 9K iterations leads to substantial improvements, while the performance saturates between 9K and 12K. This demonstrates that (i) the geometry field stage is indeed crucial for converging to high-quality HDR geometry, and (ii) the method is not overly sensitive to the exact iteration count once a sufficient number of iterations (≈9K) is used.
>
>
> Thank you for your suggestion. We have also added these experiments to the Appendix (see Sec. F and Table 4, 11).

---

> ### Author Response · Authors · 2025-11-24
> **Response to Weakness 3 for Reviewer JLWc**
>
> **Weakness3. Detailed explanation of the derivation of Expo-SDF.**
>
>
> The appendix assumes that a single Gaussian dominates locally. This assumption is only used to obtain a closed-form local quadratic approximation of the log-density, and is meant as a sufficient condition for analysis rather than a strict requirement of the method. The actual definition of the exposure-normalized density $\dot{d}(p)$ in Eq. 8 is a full mixture over all Gaussians, and the pseudo-SDF is always computed from this mixture. In other words, our algorithm does not require that exactly one Gaussian dominates at every location; the dominance assumption is only a simplifying device to show that, when it approximately holds, the log-density behaves like a Mahalanobis distance and thus justifies the SDF-like interpretation.
>
>
> When several Gaussians contribute comparable densities, the pseudo-SDF remains well behaved for three reasons:
>
>
> **1.** $\dot{d}(p)$ is the smooth sum of exposure-weighted Gaussian distributions. Its negative logarithm $-2 \log \dot{d}(p)$ defines a smooth potential energy, which can be regarded as a soft minimization of multiple quadratic forms. As a result, Eq. 9 changes continuously as the relative contributions of nearby Gaussians vary; there is no hard switch or discontinuity when the dominant Gaussian changes.
>
>
> **2.** The choice of $g^*$ affects the local scaling factor, but the overall shape of the field is governed by $-2 \log \dot{d}(p)$, which already aggregates all nearby Gaussians. In densely overlapping or transparent regions, the behavior of the pseudo-SDF also resembles that of mixture distance fields in traditional SDF methods, which have been empirically shown to be robust under multiple Gaussian overlaps.
>
> **3.** Eqs. 7-9 automatically downweight Gaussians originating from severely overexposed or underexposed regions. In areas where multiple Gaussian distributions overlap but exhibit differing exposure reliability, this exposure-aware weighting effectively clarifies the mixture, restores discernible quasi-dominant contributions, and stabilizes the near-surface pseudo-SDF.
>
>
> We greatly appreciate your constructive feedback and continued support.

---

### Official Review · Reviewer_v1b4 · 2025-10-29

**Soundness:** 2
**Presentation:** 3
**Contribution:** 2
**Rating:** 4
**Confidence:** 5

**Summary:**

The paper Expo-GS introduces a new framework for High Dynamic Range Novel View Synthesis (HDR-NVS) by incorporating an exposure-aware geometric module into the 3D Gaussian Splatting (3D-GS) pipeline. The approach decomposes the task into color, geometry, and exposure components, inspired by the mechanisms of human visual perception. This design elegantly tackles a major limitation in previous methods. The proposed Exposure-Aware Signed Distance Function (Expo-SDF) makes a notable technical contribution by improving the robustness of geometry learning under complex lighting conditions.

**Strengths:**

1. The paper presents a comprehensive experimental evaluation, demonstrating state-of-the-art performance on standard HDR-NVS benchmarks and providing thorough ablation studies to validate the contribution of its components.

2. The proposed Expo-SDF module is a well-executed integration of an exposure-weighting mechanism into an SDF framework; however, this combination feels more like a straightforward and incremental assembly of existing concepts rather than a fundamental architectural innovation.

3. The paper is well-written and the figure is clear.

**Weaknesses:**

1. Overall, this paper presents an effective approach that combines HDR rendering with an exposure-aware SDF. However, the modular and stage-wise nature of its technical design makes the work read more like a well-engineered integration of existing techniques than a fundamentally innovative contribution.

In particular, the core component, Expo-SDF, is essentially an intuitive exposure-weighted extension of the traditional SDF, with limited conceptual novelty. While the paper successfully demonstrates that combining HDR and SDF is effective, it does not provide sufficient justification that such a combination is necessary or deeply integrated at the algorithmic level. Consequently, the contribution of this work lies more in its solid engineering implementation and practical system design than in proposing a genuinely new research direction or model architecture with substantial theoretical insight.


2. The comparison methods are too limited. Basically, only HDR-NeRF and HDR-GS among the compared methods are in a similar direction. The lack of sufficient comparative experiments makes the results less convincing. Perhaps some related methods, such as those in (1)–(4), could also be included for comparison. The authors’ expertise in this area is also questionable.

(1). High Dynamic Range Novel View Synthesis with Single Exposure (ICML 2025)

(2). HDR-HexPlane: Fast High Dynamic Range Radiance Fields for Dynamic Scenes (3DV 2024)

(3). LTM-NeRF: Embedding 3D Local Tone Mapping in HDR Neural Radiance Field (TPAMI 2024), this work is an extension of HDR-NeRF.

(4). HDR-Plenoxels: Self-Calibrating High Dynamic Range Radiance Fields (ECCV 2022)


3. The paper does not include a dedicated Limitations section. Although some constraints can be inferred—such as computational overhead and dependence on multi-exposure data, explicitly acknowledging them would enhance the paper’s clarity and transparency.

**Questions:**

The paper thoroughly compares against HDR-GS but could be even stronger by including a direct ablation or comparison against a "vanilla" SDF integrated into 3D-GS (without the exposure-aware component). Table 6 compares to other SDF methods, but a controlled ablation within the Expo-GS framework (e.g., "Expo-GS w/ standard SDF loss") would more directly isolate the contribution of the exposure-aware part of the Expo-SDF.

---

> ### Author Response · Authors · 2025-11-24
> **Response to Weaknesses 1 and 2 for Reviewer v1b4**
>
> We appreciate your comments and are grateful for the positive feedback on our geometry field design, technical contributions, and the clarity of our writing and figures. For your questions, we provide further clarifications and additional justifications below.
>
> ### **Weakness1. Clarifying the core concept of our method.**
>
> Previous HDR-NVS methods tended to **overemphasize color** while underestimating the roles of geometry and exposure. They primarily focused on irradiance representations and tone-mapping modules, both of which are optimized directly from color residuals. In contrast, our method, inspired by the **human visual system (**HVS**)**, elevates HDR-NVS from a “**color-only**” paradigm to a unified “**color + geometry + exposure**” formulation and introduces a novel exposure-aware Expo-SDF. Experimental results show that our method effectively alleviates issues inherent in multi-exposure HDR systems, such as unreliable supervision in overexposed and underexposed regions. These issues typically lead to **geometric drift, boundary blurring, and inconsistencies** between color and geometry.
>
> Specifically, our method contributes along three axes:
>
> ### **1.** We have proposed a novel HVS-inspired paradigm for HDR-NVS.
>
> Inspired by the functional organization of the HVS, Expo-GS decomposes HDR-NVS into color, geometry, and exposure components, introduces an exposure-aware signed distance function that reweights geometric supervision based on local exposure and structural reliability, and adopts a disentangled-then-joint training paradigm that mirrors the dual-pathway processing and integration of color and structure in HVS.
>
> ### **2.** We have proposed a novel mathematical representation.
>
> Eqs.(7)-(10) jointly define a new Expo-SDF via the per-view exposure reliability and the exposure-normalized density. This representation is not only distinct from traditional SDFs but also departs from the color-driven paradigm that dominates existing HDR-NVS methods: instead of being derived from color statistics, it induces a new geometric field grounded in HDR-specific exposure characteristics.
>
> ### **3.** We have proposed a new control signal for the entire pipeline.
>
> Expo-SDF is not a simple auxiliary loss term. It actively controls the structural dynamics of 3D-GS: it guides Gaussian growth (Eq. 13), triggers Gaussian pruning (Eq. 14), and participates in joint color–geometry optimization (Eq. 15). In this sense, Expo-SDF is not merely additional supervision, but the core control field that determines how the 3D-GS representation is updated over time.
>
> This is the **first work** to unify multi-exposure irradiance modeling and geometric modeling within the GS framework for HDR-NVS tasks, explicitly leveraging exposure reliability to regulate geometric supervision and point cloud growth/pruning.
>
> ### **Weakness2. Provide more comparisons with additional methods.**
>
> We selected HDR-NeRF and HDR-GS as baselines because they are widely used representatives of the NeRF-based and GS-based families, respectively. However, we agree that including additional comparison methods can further strengthen the empirical evaluation, and the corresponding results are summarized below.
>
> | Method | | **LDR-OE** | | | **LDR-NE** | | | **HDR** | |
> |:-------|:-----:|:-----:|:------:|:-----:|:-----:|:------:|:-----:|:-----:|:------:|
> | | PSNR↑ | SSIM↑ | LPIPS↓ | PSNR↑ | SSIM↑ | LPIPS↓ | PSNR↑ | SSIM↑ | LPIPS↓ |
> | GaussHDR | 39.74 | 0.974 | 0.021 | 36.89 | 0.977 | 0.018 | 37.66 | 0.967 | 0.025 |
> | Mono-HDR | 30.16 | 0.923 | 0.051 | 31.44 | 0.938 | 0.055 | 38.81 | 0.975 | 0.014 |
> | LTM-NeRF | 39.42 | 0.976 | 0.033 | 36.67 | 0.964 | 0.027 | 36.55 | 0.953 | 0.018 |
> | HDR-Plenoxels | 36.38 | 0.959 | 0.045 | 35.02 | 0.943 | 0.045 | 33.86 | 0.941 | 0.046 |
> | **Expo-GS (Ours)** | **41.38** | **0.989** | **0.010** | **37.47** | **0.984** | **0.014** | **39.06** | **0.981** | **0.010** |
>
>
>
> All comparison methods are based on the training and testing set partitions declared in Section 4.1 Experimental Settings of our paper. Experimental results demonstrate that our method still achieves a significant lead, outperforming additional methods in both overall reconstruction quality and perceptual consistency across LDR-OE, LDR-NE, and HDR settings.
> All the methods you mentioned have already been cited. We have also added this experiments to the Appendix (see Sec. F and Table 6).

---

> ### Author Response · Authors · 2025-11-24
> **Response to Weakness 3 and Question 1 for Reviewer v1b4**
>
> ### **Weakness 3. Supplementing the Discussion on Limitations.**
>
> | **Method** | **Training time (min)** | **GPU memory (GB)** | **Inference speed (fps)** |
> |:-----------|:-----------------------:|:-------------------:|:-------------------------:|
> | HDR-NeRF | 517 | 80 | 0.128 |
> | HDR-GS | 33 | 8 | 122 |
> | Expo-GS (ours) | 36 (8 / 21 / 7) | 6 / 11 / 9 | 131 |
>
>
> Although Expo-GS performs well on HDR-NVS, it also has limitations. Our joint optimization introduces additional training overhead because of the multi-stage schedule and the Expo-SDF refinement. The moderate increase in training time mainly comes from the second stage (Geometry Field Training). However, when combined with the single-pass rendering strategy in the first stage and the lightweight optimization in the third stage, the overall training time remains reasonable, and our method still outperforms other methods in terms of inference speed. As shown by the main HDR-NVS experiments, our method achieves a favorable balance between performance and efficiency, where “(8 / 21 / 7)” denotes the relative training cost of the three stages.
>
>
> Additionally, the reliance on multi-exposure inputs requires strictly static scenes, which limits applicability to dynamic settings. Future work will explore extensions to time-varying scenes in real-world environments.
> Thank you for your suggestion. We have added the corresponding limitations section in Appendix A.
>
> ### **Question 1. Integrate standard SDF for comparison.**
>
> We agree that a vanilla SDF variant can enhance the transparency of our method and highlight its effectiveness. We conducted supplementary experiments as shown in the table above (with vanilla SDF):
>
>
> | Method | | **LDR-OE**  | | | **LDR-NE**  | | | **HDR** | |
> |:-------|:-----:|:-----:|:------:|:-----:|:-----:|:------:|:-----:|:-----:|:------:|
> | | PSNR↑ | SSIM↑ | LPIPS↓ | PSNR↑ | SSIM↑ | LPIPS↓ | PSNR↑ | SSIM↑ | LPIPS↓ |
> | Expo-GS (w/ Vanilla SDF) | 36.59 | 0.954 | 0.038 | 34.36 | 0.948 | 0.044 | 32.95 | 0.947 | 0.046 |
> | **Expo-GS (w/ Expo-SDF)** | **41.38** | **0.989** | **0.010** | **37.47** | **0.984** | **0.014** | **39.06** | **0.981** | **0.010** |
>
>
> The experiments demonstrated that the vanilla SDF fails to deliver satisfactory results and cannot reasonably handle the exposure conditions of HDR-NVS. We attribute this to the fact that the vanilla SDF forced constraints fail to properly process exposure information, leading to field geometry conflicts. This causes conflicts between irradiance field and geometry field especially in the overexposed and underexposed regions. This experiment further demonstrates the effectiveness of Expo-SDF.
> Thank you for your suggestion. We have also added this experiments to the appendix (see Sec. F and Table 7).
>
>
> We greatly appreciate your constructive feedback and continued support.

---

### Official Review · Reviewer_PFsF · 2025-10-31

**Soundness:** 4
**Presentation:** 4
**Contribution:** 3
**Rating:** 6
**Confidence:** 4

**Summary:**

The paper presents Expo-GS, a High Dynamic Range (HDR) novel view synthesis framework. The paper claims to remove radiometric bias and geometric inconsistency caused by varying exposures. A comprehensive qualitative and quantitative evaluation follows.

**Strengths:**

1. The paper is well written and clear.
2. The methodology is sound and the motivation to design a framework like this is great! This seems to be the first method to jointly model geometry and irradiance for 3D reconstruction.
3. The results presented are impressive and quantitative evaluations seems to support the same.

**Weaknesses:**

1. No supplementary videos to verify the quality of the 3D reconstruction.

**Questions:**

1. Please cite PyTorch.

2. I would be willing to raise my rating if the 3D scene reconstruction videos are also provided as supplementary/shown as a sequence of images to further verify the fidelity of reconstructions.

---

> ### Author Response · Authors · 2025-11-24
> **Response to Reviewer PFsF**
>
> We appreciate your thoughtful comments and are grateful for the positive feedback regarding the soundness of our methodology and  the clarity of our writing. Thank you for your time and constructive review. Below we provide a point-by-point response to your comments.
>
>
> ### **Weakness 1 and Question 2. Provide reconstruction videos.**
> Thank you for the suggestion. We have added the reconstruction video to the supplementary material and kindly invite you to review it. The video further demonstrates that our method achieves excellent color fidelity and structural consistency.
>
>
> ### **Question 1. Add relevant references.**
> Thank you for the reminder. We have added the citation for PyTorch in the revised paper (927 Lines).
>
>
> We greatly appreciate your constructive feedback and continued support.

---

> > ### Comment · Reviewer_PFsF · 2025-11-26
> >
> > The videos of the reconstructions look pretty good but they are only on synthetic scenes.
> >
> > All my queries have been answered.
> >
> > I'm happy with my current rating - 6

---

> > > ### Author Response · Authors · 2025-11-28
> > > **Response to real-world reconstruction videos**
> > >
> > > Thank you very much for your thoughtful and detailed response.
> > >
> > > We have included additional reconstruction videos of real-world scenes in the updated supplementary materials.
> > >
> > > We sincerely appreciate your time and valuable feedback and wish you a wonderful day.

---

### Official Review · Reviewer_6ACF · 2025-10-31

**Soundness:** 2
**Presentation:** 2
**Contribution:** 3
**Rating:** 4
**Confidence:** 4

**Summary:**

This paper introduces a new framework for high-dynamic-range novel view synthesis (HDR-NVS). Specifically, HDR-NVS is decomposed into three stages: radiance field training, geometry training, and joint training. The key idea of this paper is to estimate the exposure reliability to modulate the geometry by lowering the weights for unstable regions. Besides, this paper also achieves exposure-aware density control in the joint control stage. Experiments are conducted to show the effectiveness of the proposed Expo-GS method.

**Strengths:**

- It's a good idea to consider the exposure reliability when reconstructing the geometry.
- This paper achieves superior performance, especially on real-world datasets.

**Weaknesses:**

- Eqs. 7-8 define an exposure score and apply an inverse weighting.
  This necessarily down-weights saturated regions but up-weights under-exposed, low-SNR areas, conflicting with the claim of suppressing both.
  The aggregation from the per-view quantity in Eq. 7 to the global quantity used in Eq. 8 is under-specified, leaving ambiguity in how per-view scores affect the 3D density.

- Eq. 9 uses a hard nearest-Gaussian/min-axis selection to construct the SDF surrogate.
  This introduces non-smoothness: gradients can switch discontinuously as the winning Gaussian/axis changes, especially near surface boundaries.
  Such discontinuities propagate to the normal consistency in Eq. 11 and to the growth/pruning triggers in Eqs. 13--14, risking training instability.

- Compared baselines are not extensive, especially lacking several recently published HDR methods (GaussHDR (CVPR'25), Mono-HDR-3D (ICML'25)).

- In Figure 4, only 3D-GS is compared, while visualizations of some SOTA baselines like HDR-NeRF and HDR-GS are provided.
  More importantly, there is a mislabeling in Table 2: HDR-NeRF performs better than the proposed Expo-GS in the LDR-NE ($t_2$, $t_4$) setting (PSNR).

- No limitations and failure cases are discussed in this paper.

**Questions:**

I choose to give a score of 4 because I think a score of 2 is unfair to this paper. On the other hand, I need to kindly remind the authors that the majority of my questions raised in the weakness part should be addressed to maintain this rating. Of course, it is still possible for me to improve my rating if the authors well address all my concerns. For detailed questions, please see the weakness part.

---

> ### Author Response · Authors · 2025-11-24
> **Response to Weaknesses 1 and 2 for Reviewer 6ACF**
>
> We appreciate your thoughtful comments and are grateful for the positive feedback on our core idea of designing exposure reliability for geometry field reconstruction, as well as your recognition that our method achieves superior performance. For the identified weaknesses, we provide further clarifications and additional experiments below.
>
> ### **Weakness1. Detailed Explanation of Eq. 7 and Eq. 8.**
>
> We would like to clarify that the inverse weighting in Eq. 7 and Eq. 8 does not up-weight underexposed regions. In Eq. (8), the effective contribution of the j-th Gaussian to the Expo-SDF field is proportional to $\displaystyle \frac{\alpha_j}{\mathcal{E}(\mu_j) + \epsilon}$. Whether a region is up-weighted or down-weighted therefore depends jointly on ${\alpha_j}$ and $\mathcal{E}(\mu_j) + \epsilon$, rather than on $\mathcal{E}(\mu_j)$ alone.
>
>
> In overexposed regions, $\mathcal{E}(\mu_j)$ becomes large, so it appears in the denominator and strongly attenuates the contribution of the corresponding Gaussians. In underexposed regions, however, ${\alpha_j}$ becomes very small: in regions with extreme underexposure, the low-signal conditions lead to unstable gradients, and the learned Gaussian opacities tend to be close to zero. As a result, the ratio of $\displaystyle \frac{\alpha_j}{\mathcal{E}(\mu_j) + \epsilon}$ becomes very small (with $\epsilon > 0$ fixed), so the contribution of underexposed Gaussians is also effectively suppressed.
>
>
> Simultaneously, the constant $\epsilon$ is introduced for numerical stability, to avoid division-by-zero when $\mathcal{E}(\mu_j)$ is close to zero, and it does not introduce any additional up-weighting of underexposed regions. Overall, Eq. 7 and Eq. 8 do not preferentially up-weight underexposed areas; instead, both extreme exposure regimes (overexposed and underexposed) are controllably attenuated through different terms within the same formulation.
>
>
> The exposure estimate for each training view is first obtained by sampling at the projection of each Gaussian center in that view. These per-view estimates are then aggregated into a single per-Gaussian exposure $\mathcal{E}(\mu_j)$ via a visibility-weighted average over all views in which the Gaussian is visible. This procedure directly parallels how 3D-GS accumulates per-view signals into per-Gaussian statistics (e.g., SH color and opacity) through its differentiable rasterization pipeline. We apply the same standard per-view to per-Gaussian aggregation to exposure, so the global quantity in Eq. 8 is well defined rather than ambiguous.
>
>
> ### **Weakness2. Detailed explanation of Eq. 9.**
>
> The term $g^*$ may have created a misconception of hard winner selection, which was then extended to min-axis selection. In reality, $\dot{d}(p)$ is an exposure-normalized mixture of Gaussians (Eq. 8). This mixture, and hence its logarithm, is a smooth and fully differentiable function of all Gaussian parameters.
>
>
> The role of $g^*$ is only to provide a local scale and normal direction for interpreting this potential as an SDF surrogate; it does not change the underlying log-density field.
>
>
> Concretely, by “dominant Gaussian” we simply mean the Gaussian with the largest volumetric contribution in the standard front-to-back compositing used in 3D-GS. This notion of dominance arises from continuous transmittance accumulation: as densities evolve, the transmittance-weighted contributions of neighboring Gaussians vary smoothly, so no discrete argmin/argmax operator is introduced into the computational graph. Consequently, the pseudo-SDF $f_{\text{HDR}}(p)$ and its gradient $\nabla f_{\text{HDR}}(p)$ remain smooth, and the normal-consistency term in Eq. 11 inherits this smoothness as well.
>
>
> Regarding Eqs. 13-14, the growth and pruning rules use scalar criteria derived from $f_{\text{HDR}}(p)$ to duplicate or remove Gaussians. In practice, these are implemented as heuristic, non-differentiable updates interleaved with gradient steps (similar to densification/pruning in standard 3D-GS), rather than as differentiable operators in the gradient computation. Thus, they do not introduce additional non-smooth gradients, and empirically we have not observed any instability attributable to Eq. 9.

---

> ### Author Response · Authors · 2025-11-24
> **Response to Weaknesses 3, 4, and 5 for Reviewer 6ACF**
>
> ### **Weakness3. Provide more comparisons with additional methods.**
>
> Your suggestion to incorporate additional comparisons is beneficial for further enhancing the impact of our paper. Below are the corresponding comparison results.
>
>
>
> | Method | | **LDR-OE** | | | **LDR-NE** | | | **HDR** | |
> |:-------|:-----:|:-----:|:------:|:-----:|:-----:|:------:|:-----:|:-----:|:------:|
> | | PSNR↑ | SSIM↑ | LPIPS↓ | PSNR↑ | SSIM↑ | LPIPS↓ | PSNR↑ | SSIM↑ | LPIPS↓ |
> | GaussHDR | 39.74 | 0.974 | 0.021 | 36.89 | 0.977 | 0.018 | 37.66 | 0.967 | 0.025 |
> | Mono-HDR | 30.16 | 0.923 | 0.051 | 31.44 | 0.938 | 0.055 | 38.81 | 0.975 | 0.014 |
> | LTM-NeRF | 39.42 | 0.976 | 0.033 | 36.67 | 0.964 | 0.027 | 36.55 | 0.953 | 0.018 |
> | HDR-Plenoxels | 36.38 | 0.959 | 0.045 | 35.02 | 0.943 | 0.045 | 33.86 | 0.941 | 0.046 |
> | **Expo-GS (Ours)** | **41.38** | **0.989** | **0.010** | **37.47** | **0.984** | **0.014** | **39.06** | **0.981** | **0.010** |
>
>
> All comparison methods are based on the training and testing set partitions declared in Section 4.1 (Experimental Settings) of our paper. Experimental results demonstrate that our method still achieves a significant lead, outperforming additional methods in both overall reconstruction quality and perceptual consistency across LDR-OE, LDR-NE, and HDR settings.
> We have also added this experiments to the Appendix (see Sec. F and Table 6).
>
>
> ### **Weakness4. Optimize the layout of visualization comparison.**
>
> Due to page limitations, the visual comparison has been provided in the appendix as Figure 8, and the editorial errors have been corrected.
>
> ### **Weakness 5. Supplementing the Discussion on Limitations.**
>
>
> | **Method** | **Training time (min)** | **GPU memory (GB)** | **Inference speed (fps)** |
> |:-----------|:-----------------------:|:-------------------:|:-------------------------:|
> | HDR-NeRF | 517 | 80 | 0.128 |
> | HDR-GS | 33 | 8 | 122 |
> | Expo-GS (ours) | 36 (8 / 21 / 7) | 6 / 11 / 9 | 131 |
>
>
>
> Although Expo-GS performs well on HDR-NVS, it also has limitations. Our joint optimization introduces additional training overhead because of the multi-stage schedule and the Expo-SDF refinement. The moderate increase in training time mainly comes from the second stage (Geometry Field Training). However, when combined with the single-pass rendering strategy in the first stage and the lightweight optimization in the third stage, the overall training time remains reasonable, and our method still outperforms other methods in terms of inference speed. As shown by the main HDR-NVS experiments, our method achieves a favorable balance between performance and efficiency, where “(8 / 21 / 7)” denotes the relative training cost of the three stages.
>
>
> Additionally, the reliance on multi-exposure inputs requires strictly static scenes, which limits applicability to dynamic settings. Future work will explore extensions to time-varying scenes in real-world environments. Thank you for your suggestion. We have added the corresponding limitations section in Appendix A.
>
>
> We greatly appreciate your constructive feedback and continued support.

---

### Author Response · Authors · 2025-12-03

We sincerely thank the reviewers for their constructive feedback and valuable suggestions. During the rebuttal phase, we addressed each concern in detail.


Our work is inspired by the **human visual system (HVS)** and distinguishes itself from prior HDR-NVS studies through the introduction of an **exposure-aware signed distance function (Expo-SDF)**, which dynamically reweights geometric supervision based on local exposure reliability. We also propose an interactive optimization strategy to balance exposure-aware irradiance density control with accurate geometric modeling.


The reviewers highlighted several strengths:

**1. Novel idea with superior performance (Reviewer-6ACF)**

**2. Sound motivation and well-written presentation (Reviewer-PFsF)**


**3. Comprehensive experimental evaluation and clearly written (Reviewer-v1b4)**

**4. Interesting concept with solid validation (Reviewer-JLWc)**




We have addressed the main concerns as follows:


**1. Supplemented the Discussion on Limitations.**

Although Expo-GS performs well on HDR-NVS, it also has limitations. Our joint optimization introduces additional training overhead because of the multi-stage schedule and the Expo-SDF refinement. The moderate increase in training time mainly comes from the second stage (Geometry Field Training). However, when combined with the single-pass rendering strategy in the first stage and the lightweight optimization in the third stage, the overall training time remains reasonable, and our method still outperforms other methods in terms of inference speed. As shown by the main HDR-NVS experiments, our method achieves a favorable balance between performance and efficiency.

**2. Provided more comparisons with additional methods.**


Experimental results demonstrate that our method still achieves a significant lead, outperforming additional methods in both overall reconstruction quality and perceptual consistency across LDR-OE, LDR-NE, and HDR settings. We have also added these experiments to the Appendix (see Sec. F and Table 6).

To our knowledge, this is the **first work** to leverage SDF into HDR-NVS, supported by a thorough theoretical foundation and analysis. Our method offers new insights into exposure-aware scene reconstruction and novel view synthesis.

We hope the AC and reviewers find our responses satisfactory. Thank you for your time and effort.

---

### Meta-Review · Area_Chair_g6vd · 2026-01-02

**Summary:**

This paper proposes Expo-GS, an exposure-aware extension of Gaussian Splatting for high dynamic range novel view synthesis, introducing an exposure-weighted SDF and a multi-stage optimization pipeline. Reviewers generally find the motivation reasonable and the empirical results competitive, particularly on HDR benchmarks. However, the core contribution is viewed as incremental, largely building on existing GS and SDF formulations with exposure-aware reweighting, and falling short of ICLR. Despite extensive rebuttal clarifications and additional experiments, the overall assessment remains unchanged. Overall, the work is solid but better suited to a vision-focused venue such as ECCV. The AC recommends rejection.

**Reviewer Concerns:**

Reviewer concerns focus on the depth of the contribution, its novelty, and its positioning. While incorporating exposure reliability into geometric supervision is sensible and yields performance gains, reviewers consistently view Expo-SDF as an intuitive extension rather than a fundamentally new modeling approach. Questions remain about the necessity and generality of the three-stage training pipeline, the method's sensitivity to design choices, and whether the method offers new insights beyond improved engineering and tuning. Comparisons to related HDR-NVS methods are expanded in the rebuttal, but concerns about limited conceptual novelty and scope remain. Overall, the rebuttal strengthens the empirical case but does not alter the conclusion that the contribution is incremental.

**Reviewer Scores:**

- Reviewer 6ACF: Scores the paper 4. While the motivation and performance are solid, concerns about formulation clarity, baseline coverage, and novelty remain after rebuttal.
- Reviewer PFsF: Scores the paper 6. The rebuttal addresses the reviewer’s questions, and the assessment remains unchanged at a borderline accept level.
- Reviewer v1b4: Scores the paper 4. The work is viewed as a well-engineered combination of existing ideas rather than a substantive algorithmic advance, and this assessment remains unchanged.
- Reviewer JLWc: Scores the paper 4. Concerns about generalization and justification of key assumptions persist, keeping the assessment below the acceptance threshold.

---

### Decision · Program_Chairs · 2026-01-26

Reject